



# STEAD: A high-resolution daily gridded temperature dataset for Spain

*Roberto Serrano-Notivoli[1], Santiago Beguería[1] and Martín de Luis[2]

[1] Estación Experimental de Aula Dei (EEAD-CSIC), Zaragoza, E-50059, Spain.
[2] Department of Geography and Regional Planning and Environmental Sciences Institute (IUCA), University of Zaragoza, Zaragoza, E-50009, Spain.

*Correspondence to*: Roberto Serrano-Notivoli (rserrano@eead.csic.es)

**Abstract.** Using the full total of available 5,520 observatories covering the whole territory of Spain, a daily gridded maximum and minimum temperature was built covering a period from 1901 to 2014 in peninsular Spain and 1971-2014 in Balearic and Canary Islands. A comprehensive quality control was applied to the original data and the gaps were filled on each day and location independently. Using the filled data series, a grid of 5 x 5 km spatial resolution was created by estimating daily temperatures and their corresponding uncertainties at each grid point. Four daily temperature indices were calculated to describe the spatial distribution of absolute maximum and minimum temperature, number of frost days and number of summer days in Spain. The southern plateau showed the maximum values of maximum absolute temperature and summer days, while the minimum absolute temperature and frost days reached their maximums at northern plateau. The use of all the available information, the complete quality control and the high spatial resolution of the grid allowed for an accurate estimate of temperature that represents a precise spatial and temporal distribution of daily temperatures in Spain. STEAD dataset is publicly available at http://dx.doi.org/10.20350/digitalCSIC/8622 and can be cited as Serrano-Notivoli et al. (2019).

## 1 Introduction

Despite a clear improvement over the last decades in meteorological measurement techniques, the inclusion of automated systems with near-real-time information submission, or the increasing number of stations with a growing number of recorded variables, the existing climatic information is still unrepresentative in many territories. The low density of stations in isolated areas and the great variability in the number and location of observations over time represent a substantial problem. Despite these problems, or perhaps due to them, different team dedicated great effort to creating reliable gridded climatic datasets covering large time periods. Between all the climate variables, temperature datasets are among the most popular, such as the CRUTEM (1850-2017) (Jones et al., 2010), Willmott and Matsura (1900-2014) (2001), WorldClim (1970-2000) (Fick and Hijmans, 2017), GISTEMP (1880-2017) (Hansen et al., 2010), or BEST (1850-2017) (Rohde et al., 2013). In this regard, temperature has been also widely studied in Spain, in terms of its spatio-temporal distribution (e.g. Peña-Angulo et al., 2016) and temporal trends (e.g. González-Hidalgo et al., 2015 and 2018). Nevertheless, most of the existing works addressed coarse temporal scales or used individual stations for detailed regions (e.g. Villeta et al., 2018).



Monthly gridded datasets have enabled a better understanding of the climatic dynamics of the planet, especially as an element of quantification and validation of climate change, due to their ability to reproduce the mid-frequency variability of temperature. However, most of the methodologies used in those works are not suitable for addressing the variability of temperature at the daily scale, due to the higher spatial and temporal variability and because of the larger number of input

stations required to build a reliable dataset. Although climate change in respect to temperature is often quantified in terms of changes in its mean values, most of the risks attributed to climate change are, however, related to temperature extremes that occur at shorter time scales, such as the daily scale. The study of climate change signals in temperature extremes is therefore still largely pending due to the absence of reliable daily datasets representative of most of the territories. This absence of daily gridded datasets, with several remarkable exceptions (e.g. Cornes et al., 2018; Lussana et al., 2018), is linked to: i) an absence

of contrasted reconstruction methodologies owing to the different temporal and spatial structure of daily data in contrast to monthly or annual values; and ii) an absence of contrasted quality control protocols for daily time series.

In addition, the reliability of a dataset not only depends on the resolution but on the consistency of the data. Quality control processes are crucial to create trustworthy datasets and, although the many existing approaches (e.g.: Haylock et al., 2008; Klein-Tank et al., 2002, Klok and Klein-Tank, 2009) apply basic procedures, some others go beyond and check for spatial

consistency (e.g.: Lussana et al., 2018; Feng et al., 2004), which is recommended when using high-density networks.

The same problems intervene, though more severely, in global datasets. At the sub-regional and local scales, the understanding of high-resolution climatic variability is of key importance in a context of global change, and these datasets often are not adequate to address specific research questions such as extremes or small variations affecting other components of the natural system, due to a low spatial or temporal resolution.

Daily scale in temperature information is of key importance in many areas. The E-OBS dataset (Cornes et al., 2018) is the best example of daily gridded dataset for large international areas thanks to the integration of thousands of transboundary climate data. However, it does not pretend to be comprehensive for specific regions (Van Den Besselaar et al., 2015) and a deeper analysis with more information is required for higher spatial scales. The Spanish territory perfectly captures the great climatic variability with very different regimes in a small area that leads to high risks related to potential changes of this variability.

Currently, there are only two daily gridded datasets available for Spain: E-OBS (the Spanish part of the European dataset) and Spain02 (Herrera et al., 2016). Although both of them have been checked for their reliability, and are useful for specific purposes, they have limitations that prevent several climatic analyses. For instance, in their construction they did not considered all the available information but only a few stations as basis for creating the grid (229 and 250, respectively), prioritizing the longest data series over a higher spatial density. This approach is suitable for wide-ranging temperature studies, yet insufficient

when addressing small areas with great variability.

This article introduces the STEAD (Spanish TEmperature At Daily scale) dataset, a new high-resolution daily gridded (maximum and minimum) temperature dataset for Spain covering the period 1901-2014 for peninsular Spain and 1971-2014 for Balearic and Canary Islands. Based on the available quality-controlled temperature information in Spain (more than 5,000 stations), we used the same spatial resolution as SPREAD (Serrano-Notivoli et al., 2017a), its corresponding precipitation





dataset. We propose: 1) a methodology for an exhaustive quality control; and 2) a reconstruction methodology using all the available information and based on local regression models.

Section 2 describes the input data and section 3 explains the methodology used to apply the quality control, fill the gaps in the original series and the gridding process. Section 4 shows the results of the method applied to the Spanish temperature network

as well as the validation of the reconstruction and gridding procedures. Also, a brief description of four climatic indices based on daily temperature is shown. Results are discussed in section 5 and summarized in the conclusions at section 7 after the specification of the availability of the dataset in section 6.

## 2 Input data

We used the full total of available 5,520 observatories covering the whole territory of Spain, which was divided in three areas

to compute the grid: 1) Peninsular Spain (492,175 km2) with 5,056 stations covering the period 1901-2014; 2) Balearic Islands (4,992 km2), with 124 stations covering 1971-2014 and 3) Canary Islands (7,493 km2) covered 1971-2014, using 340 stations (Figure 1 down). The data sourced from the Spanish Meteorological Agency (Aemet) and from the Spanish Ministry of Environment and Agriculture (MAGRAMA).

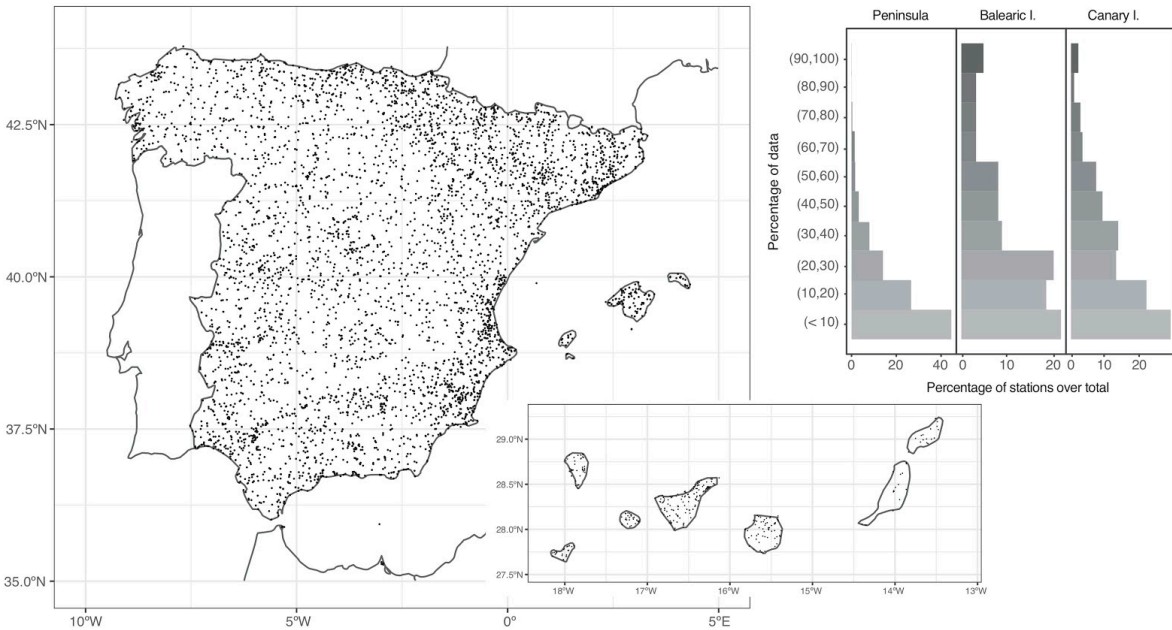

**Figure 1: Location of the temperature stations and categorisation by percentage of recorded data.**

The mean number of stations per year increased all over the studied period (Figure 2b). The first years of the 20[th] century had only a few stations available (Brunet et al., 2006; González-Hidalgo et al., 2015) with a great distance between them. Then, the number increases with the break of the Civil War (1936 – 1939) until the decade of the 1990s. Until this moment, all the



information sourced from Aemet and from that, MAGRAMA stations began to register data until the end of the period (2014). As noted in González-Hidalgo et al. (2015), the mean distance between stations barely changed from middle century as well as their mean elevation (between 500 and 550 m a.s.l.). Before that, the mean altitude experimented hard changes due to the removing or relocation of existing stations and new incorporations.

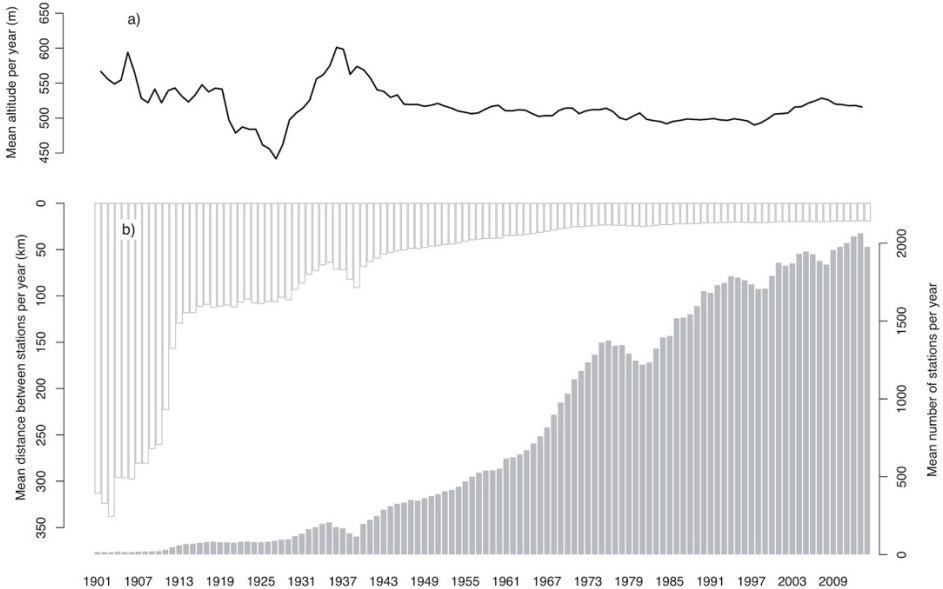

**Figure 2: Mean altitude of the stations by year (a), mean distance between stations (b, left axis, hollow bars) and mean number of stations per year (b, right axis, solid bars).**

## 3 Methods

10 The first stage is a quality control of the original dataset to remove the most obvious wrong data. From this starting point the process (Figure 3) is based on the computation of reference values (RV), which are computed for each location and day and then compared with the original values to assess the quality of the data. After a process of standardization, new values are estimated for those days without observations (or removed in the quality control process) to obtain serially-complete data series. In a final stage, the complete series are the basis to create new data series for specific pairs of coordinates that may or

15 not form a regular grid, including a measure of uncertainty for each location and day.



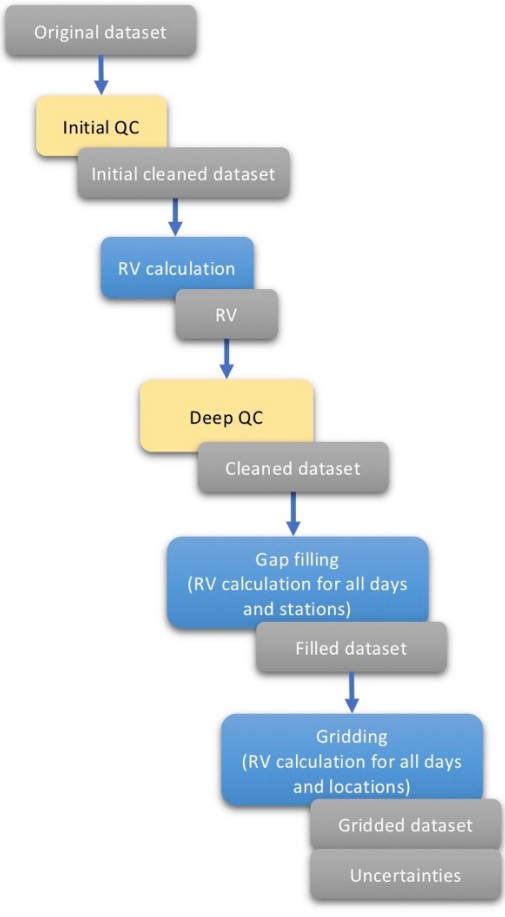

**Figure 3. Methodological protocol in a nutshell. Grey boxes represent products; yellow boxes are correction processes; and blue boxes involve RV calculations.**

### 3.1 Initial quality control

5  The *initial quality control* (*iQC*) includes five basic criteria: i) Internal coherence, ii) removal of months containing less than 3 days of data, iii) the removal of those days out of range considering: $Tmax >= 50\ ºC$ or $Tmax <= -30\ ºC$ and $Tmin >= 40\ ºC$ or $Tmin <= -35\ ºC$; iv) removal of all days in a month with a standard deviation equal to zero (suspected repeated values in the series); and v) removal of all days in a month if the sum of the differences between maximum and minimum temperatures is equal to zero (suspected duplicated values in TMAX and TMIN).

10 ### 3.2 Reference values (RV) as key process for quality control and reconstruction

Further steps in the quality control and the reconstruction process are based on the computation of reference values (RV). RV are obtained by using a *k-nearest* neighbors regression approach, which is applied to maximum and minimum temperature and to each day and location independently. The RV are estimated through the combined use of Generalized Linear Mixed Models



(GLMM) and Generalized Linear Models (GLM). The predictors (independent variables) were latitude, longitude, altitude and distance to the coast. The inclusion of these four independent variables and the building of independent models for each site and day considering only the neighborhood of the site of interest allows for large flexibility and enables capturing local features that may not be captured using other methods which result in larger spatial and temporal regularization.

5   The methodological procedure is as follows: i) *Rough Monthly RV*, which are average monthly estimates (i.e. a climatology), obtained using GLMM and all the available data; ii) then, *Fine Monthly RV*, which are monthly time series of temperature, computed using GLM and data from only the 15 nearest neighbours, and including the Rough Monthly RV obtained in the previous stage as a covariate; iii) finally, *Daily RV* are computed using GLM and data from the 15 nearest daily observations, plus the *Fine Monthly RV* of the corresponding month as added covariate. The whole process is explained in detail in the

10  following sections.



### 3.2.1. Rough Monthly RV (rmRV)

Monthly time series of daily temperature means and standard deviations were computed. Only the months with complete daily observations were used to fit the model. We used latitude, longitude, altitude and distance to the coast as fixed factors, and the year as random factor. The introduction of the year as random factor allows for isolating the fact that one particular year might be colder or warmer than the average on the whole dataset, and thus eliminates the random variability arising from the fact that the time period with observed data changes from station to station. The model, fitted independently for each month of the year and for each one of the four dependent variables defined above, can be represented as:

$$\begin{aligned} \boldsymbol{y} &\sim \mathcal{N}(\boldsymbol{X\beta} + \boldsymbol{Zb}, \ \sigma^2) \\ \boldsymbol{b} &\sim \mathcal{N}(\boldsymbol{0}, \ \boldsymbol{G}) \end{aligned} \tag{1}$$

where $\boldsymbol{y}$ is a known vector of observations with mean $E(\boldsymbol{y}) = \boldsymbol{X\beta}$ and variance $var(\boldsymbol{y}) = \sigma^2$; $\boldsymbol{\beta}$ is an unknown vector of fixed effects; $b$ is an unknown vector of random effects, with mean $E(\boldsymbol{b}) = \boldsymbol{0}$ and variance-covariance matrix $var(\boldsymbol{b}) = \boldsymbol{G}$; and $\boldsymbol{X}$ and $\boldsymbol{Z}$ are known model matrices containing the values of the fixed and random variables for the observations $\boldsymbol{y}$. The models were fit by the maximum likelihood method using the R package *lme4* (Bates et al., 2015).

Once the model parameters $\boldsymbol{\beta}, \boldsymbol{b}$ and $\sigma$ are obtained, best linear unbiased predictions (BLUPs) of the mean and standard deviation of daily temperature are calculated for each station (*i*), year (*y*) and month (*m*). At this stage a global model is fit, since all the data are used to fit the model and therefore the coefficients are assumed to be constant in space, an assumption that it is a rough simplification of reality. On the other hand, this configuration allowed us to include all the data for estimating the random year effect. The obtained estimates of mean and standard deviation for maximum and minimum temperature represent highly spatially regularized patterns of monthly temperature, and do not consider local spatial variability in the influence of the covariates.

### 3.2.2. Fine Monthly RV (fmRV)

In a second stage, monthly time series of the mean and standard deviation of daily temperature were computed again, but using a local (*k-nearest* neighbors) regression approach. For each station, the model was fit to data from the 15 nearest observations of each month plus the *rmRV* values calculated in the previous step. Inclusion of the latter as if they were legitimate observations incorporates a certain amount of spatial regularization that helps alleviating a problem that may arise when using a purely local regression approach, i.e. an excess of spatial variability, especially in areas where the model extrapolates (in latitude, longitude, altitude or distance to the coast) with respect to the neighboring locations. A Generalized Linear Model was thus built for each station and month:

$$\boldsymbol{y}' \sim \mathcal{N}(\boldsymbol{X}'\boldsymbol{\beta}', \ \epsilon) \tag{2}$$



were $y'$ is the local neighbourhood dataset, including the *Rough Monthly RV* with mean $E(y') = X'\beta'$; $\beta'$ is an unknown vector of local fixed effects; $X'$ is a known model matrix containing the values of the covariates at the 15 neighbouring sites; and $\epsilon$ is an unknown random error, which in the case of the mean temperature was assumed to be normally distributed with

zero mean, $\epsilon \sim \mathcal{N}(0, \ \sigma'^2)$, and in the case of the temperature standard deviation was modelled as following a Poisson distribution, thus taking only positive values. The obtained estimates of mean and standard deviation for maximum and minimum temperature incorporate the local variability that was lacking in the estimations of the previous step.

An example of *rmRV* and *fmRV* for a specific month is shown in the supplemental (Figure S2).

### 3.2.3. Daily RV (dRV)

In a third stage, daily maximum and minimum temperatures were predicted based on the 15 nearest observations and the *fmRV* for the corresponding month, using once again a GLM with Gaussian link:

$$y'' \sim \mathcal{N}(X'\beta'', \ \sigma''^2) \tag{3}$$

where $y''$ is the local daily neighbourhood dataset, including the *fmRV* with mean $E(y'') = X''\beta''$ and variance $var(y') = \sigma''^2$; $\beta''$ is an unknown vector of daily local fixed effects; and $X'$ is a known model matrix containing the values of the covariates at the 15 neighbouring sites. The daily estimates of each station ($dRV_{i,d,m,y}$) are then standardized (4) with the *fmRV* ($fmRV\_mean_{i,m,y}$ and $fmRV\_sd_{i,m,y}$) data to keep an equivalent standard deviation as the monthly prediction:

$$dRV\_std_{i,d,m,y} = \frac{(dRV_{i,d,m,y} - fmRV\_mean_{i,m,y})}{fmRV\_sd_{i,m,y}} \tag{4}$$

### 3.3. Quality control

After the initial quality control and the RV calculation, we have the original dataset without the most obvious anomalies and an estimate for each observation. Clearly, the *iQC* is not enough to remove inconsistencies in temporal and spatial fields. Here

is presented a novel approach of an exhaustive quality control over daily temperature data based on paired comparisons between observations ($Tobs_{i,d,m,y}$) and standardized predictions ($dRV\_std_{i,d,m,y}$). All stages of this process are carried out independently for maximum and minimum temperature. What we call *deep quality control* (*dQC*) considers similarities between observations and estimates through: i) a correlation analysis between daily observations and predictions at each analyzed location, year and month and ii) a quantification on how the differences between daily observed and predicted values

(anomalies) are spatially and temporarily distributed.



The process is iterative, which means that when *dQC* finish in the first run, and the suspect detected data are removed from the original dataset, the RV are computed again over this dataset and the *dQC* runs again. This is iterated until *dQC* does not detect any suspect data.

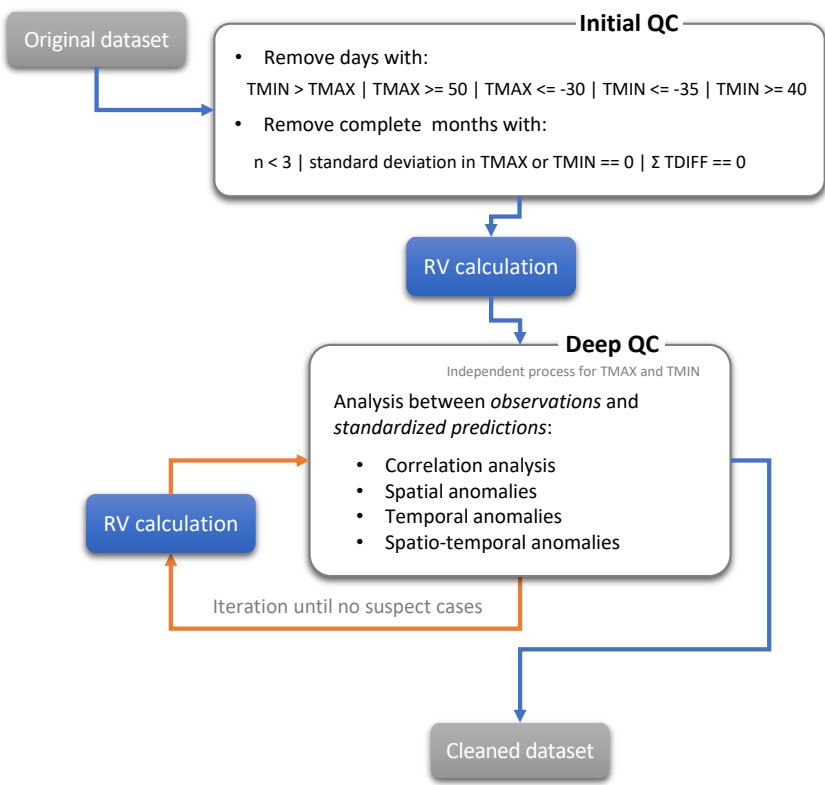

**Figure 4. Methodological protocol for quality control.**

### 3.3.1. Correlation analysis

This analysis is based on the correlation between observations and standardized predictions, and it is independent for each series of daily observations in each month. Considering a single site ($i$), month ($m$) and year ($y$) the correlation ($COR_{i,m,y}$) between observed and predicted daily data is computed and then compared with the correlation of its 15 nearest stations in the same month and year ($CORv_{i,m,y}$). The index $ZCOR$ ($ZCOR_{i,m,y}$) is then computed as the number of standard deviations that the observed correlation deviates from the observed at their neighbours (5):

$$ZCOR_{i,m,y} = \frac{\left(COR_{i,m,y} - mean(CORv_{i,m,y})\right)}{sd(CORv_{i,m,y})} \tag{5}$$

At this point, the correlations and their deviations are used to remove all daily data from the site $i$ and month $m$ if:



a) $COR_{i,m,y} \leq 0$ or,

b) $COR_{i,m,y} > 0 \,\&\, ZCOR_{i,m,y} < 0 \,\&\, p(ZCOR_{i,m,y}) < 0.001$

being $p$ the *p-value* of $ZCOR_{i,m,y}$. A negative $ZCOR_{i,m,y}$ indicates a lower observed correlation than the neighbours and $p(ZCOR_{i,m,y}) < 0.001$ indicates that it is highly unlikely that the correlation is plausible in the referred spatial and temporal context.

This part of the quality control procedure aims to detect, amongst others, (partially) repeated sequences of data, duplicated months or sequences in consecutive years, shifted dates in series or, for instance, sequences of data extremely abnormal in their spatial context. All of these anomalies, which are hard to detect with classical approaches, can be potentially identified in this stage.

### 3.3.2. Daily differences

Using the differences between the observations and the standardized predictions ($Tdif_{i,d,m,y} = Tobs_{i,d,m,y} - dRV\_std_{i,d,m,y}$), two types of anomalies are computed:

- Spatial anomaly: Each difference is compared with the differences of their 15 nearest stations ($Tdif\_v_{i,d,m,y}$). The index $Zdif\_spatial_{i,d,m,y}$ is then computed as the number of standard deviations that the observed difference deviates from their neighbours (6).

$$Zdif\_spatial_{i,d,m,y} = \frac{\left(Tdif_{i,d,m,y} - mean(Tdif\_v_{i,d,m,y})\right)}{sd(Tdif\_v_{i,d,m,y})} \qquad (6)$$

- Temporal anomaly: Each difference is compared with the differences in same station in the rest of the days of the same month and year ($Tdif\_t_{i,d,m,y}$). The index $Zdif\_temporal_{i,d,m,y}$ is then computed as the number of standard deviations that the observed difference deviates from the rest of the days of the month in same station (7).

$$Zdif\_temporal_{i,d,m,y} = \frac{\left(Tdif_{i,d,m,y} - mean(Tdif\_t_{i,d,m,y})\right)}{sd(Tdif\_t_{i,d,m,y})} \qquad (7)$$

The daily data from the site $i$ and month $m$ is removed if the absolute value of the mean of both spatial and temporal anomalies is higher than the value representing the probability of $\alpha < 0.0001$ (8).

$$\alpha_{abs(Zdifmean_{i,d,m,y})} < 0.0001 \qquad (8)$$

When the suspect data has been removed using the daily similarities and differences criteria, the RV are computed again and the quality control process starts over. This procedure is repeated until no suspect data is detected and removed.

### 3.4. Gap filling

Once quality control process is finished, a final set of RV are computed from the cleaned dataset for those locations and days with missing data. These RV corresponding with days without observations will fill the gaps, completing the series of original debugged observations ($T\_fill_{i,d,m,y}$).



## 3.5. Gridding and uncertainty

With the aim of obtain a gridded product (a regularly distributed set of data series over space), new RV are created for each location ($id$), month ($m$) and year ($y$) of the grid in three steps:

1) Grid $fmRV$ ($GfmRV\_mean_{id,m,y}$ and $GfmRV\_sd_{id,m,y}$) are created using the filled dataset ($T\_fill_{i,d,m,y}$);

2) Grid $dRV$ ($GdRV_{id,d,m,y}$) are created using the original filled dataset ($T\_fill_{i,d,m,y}$) and the computed mean monthly references;

3) Finally, the estimates are standardized using the standard deviation monthly references (9):

$$GdRV\_std_{id,d,m,y} = \frac{GdRV_{id,d,m,y} - GfmRV\_mean_{id,m,y}}{GfmRV\_sd_{id,m,y}} \tag{9}$$

In addition to the estimates of temperature for each grid point (in the second step of gridding process), we computed their corresponding uncertainty, which was calculated as the standard error of the difference between the predicted and the observed values of the 15 neighbours in each day and location.

## 3.6 Validation

The validation process consisted in the comparison between the observations and the estimates computed for each one of those observations. The assessment was carried out through seven statistical and graphical analyses:

i) A graphical comparison and Pearson calculation of the means of all the 5,520 stations considered in the study. Also, the 95th percentile of maximum temperature and 5th percentile of minimum temperature were considered to ascertain the accuracy of the extremes' prediction;

ii) a graphical representation of the Pearson correlation frequencies, by months, to show the agreement between observations and estimates;

iii) a graphical representation of counts of temperature values, by categories based on absolute values. This is useful to show potential biases in specific ranges of temperature;

iv) a collection of statistical tests to compare observations and estimates by altitudinal ranges using daily values. The tests

include the mean of observations (OBSm), the mean of estimates (PREDm), the mean absolute error (MAE), the mean error (ME), the ratio of means (RM) and the ratio of standard deviations (RSD);

v) same as (iv) but by months instead of elevations;

vi) a graphical representation of the count of temperature differences between observations and estimates; and

vii) a graphical representation of the temporal evolution of mean annual uncertainty.



### 3.7 Example of applications: daily spatial distribution and uncertainties of temperatures

Based on the gridded dataset created from the original data and with the described method, we computed four indices to show the potential applications of the grid: 1) the mean annual maximum value of daily maximum temperature; 2) the mean annual minimum value of daily minimum temperature; 3) the average annual count of days when daily minimum temperature is below

0 ºC (frost days); and 4) the average annual count of days when daily maximum temperature is over 25 ºC (summer days). All the indices were presented with their corresponding uncertainty estimate.

### 4 Results

### 4.1 Quality control

Although the quality control was carried out separately, it removed approximately a 7.4% of the original daily values both in

maximum and minimum temperatures (Table 1). The initial quality control process ($iQC$) removed a sum of 59 days out of range (less than 0.01% of the total) and 1,349 months (0.53%) containing 4,308 days (0.04%) that did not fulfil with the minimum standards set at the beginning (see section 3.1). Furthermore, the deep quality control ($dQC$) removed between 4.5 and 5.6% of the months and days considering the similarities between the observations and the estimates, being the number of removed data slightly higher in minimum temperatures. Most of the correlations in removed data were negative or very low

(Figure 5), which indicates that the observations were very different from the estimates built with their surrounding original values. The average correlations in removed data were negative both in maximum and minimum temperature, showing that the similarities were very low.

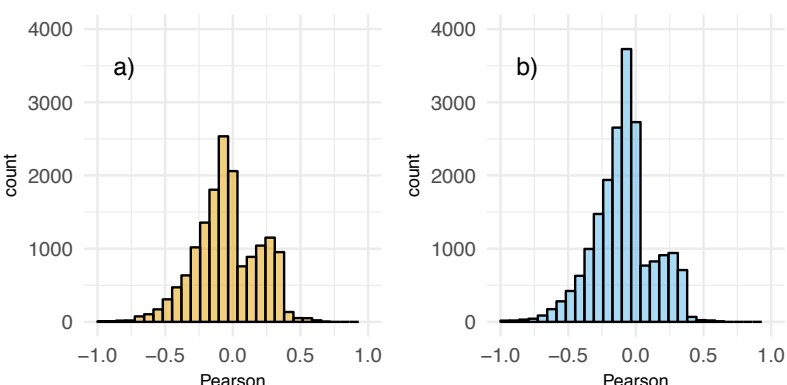

**Figure 5. Correlation frequencies between observations and estimates of removed data in quality control process. Maximum (a) and minimum (b) temperatures are shown.**

The number of removed days in maximum temperatures was higher when considering the daily spatio-temporal anomalies (Figure 6a), without a significant bias in positive nor negative differences (Figure 6b) in contrast to minimum temperatures (Figure 6e) where negative differences prevailed.



The removed values do not necessarily correspond with climatic extremes but with values that are out of the spatio-temporal context of its neighbouring observations. The fact that the maximum frequency of removed data matches with the average of maximum and minimum temperatures (Figure 6a and d) suggests that there is no bias in the suspect data detection and, indeed,

the deletions are due to errors in original data (what we intend to detect) and not to climatic extremes. Furthermore, when looking at the removed data by differences between observations and estimates (Figure 6b and e), it is noted that the maximum frequency of deletions corresponds to differences near to ±10 ºC, which is not unusual if we think that, probably, those removals are due to recording or transcription errors, related with missing decimals.

Despite the fact that the magnitudes of some of the removed data do not represent anomalous values (Figure 6c and f), they

correspond to significant anomalies in their spatial and temporal context. Beyond the magnitude in absolute terms, the differences between observations and estimates suggest, with an $\alpha < 0.0001$, that those values are very unlikely to be representative in their spatio-temporal context.

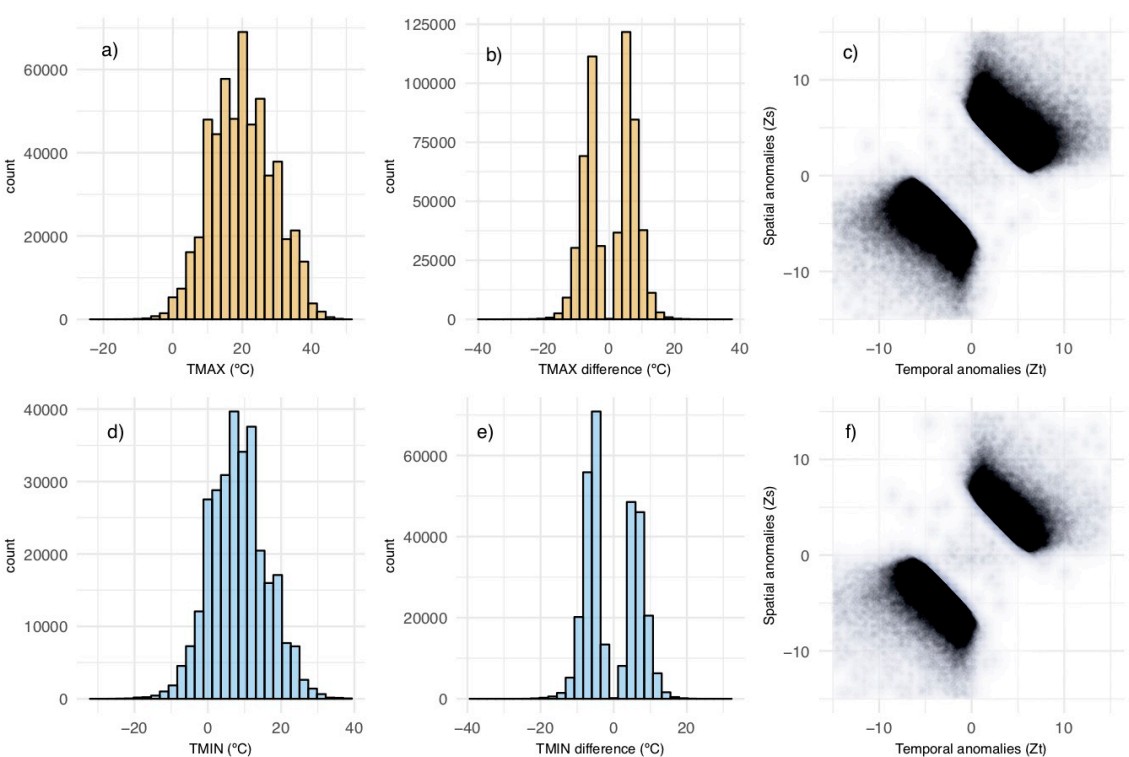

**Figure 6. Daily maximum (orange, upper line) and minimum (blue, bottom line) temperature data removed by quality control process. Left column: removed data by magnitude; central column: removed data by differences between observations and estimates; and right column: temporal anomalies (Zs) vs spatial anomalies (Zs).**

**Table 1. Number of removed days and complete months based on the quality control criteria.**



|  |  | Count |
|---|---|---|
| Daily similarities (number of removed months and days) | | |
| TMAX | Number of removed months | 15,641 (4.48%) |
|  | Number of removed days | 450,502 (4.12%) |
| TMIN | Number of removed months | 19,492 (5.58%) |
|  | Number of removed days | 566,435 (5.27%) |
| Daily differences (number of removed days) | | |
| TMAX | Number of removed days | 551,275 (3.27%) |
| TMIN | Number of removed days | 299,804 (2.09%) |

Using the reconstructed series, we built a 5x5 km spatial resolution gridded dataset of maximum and minimum temperature. The values were estimated for 1901-2014 period in peninsular Spain and for 1971-2014 period in Balearic and Canary Islands. A measure of uncertainty was added to each day and grid point of the dataset.

5   **4.2 Depurated dataset: Observations – estimates comparison**

Daily temperatures were estimated at the same location and days as the original data series but without considering the original values in each case. The comparison between the estimated temperatures and the observations showed very high correlation considering the average by stations for maximum (Figure 7a) and minimum temperatures (Figure 7c) (Pearson correlation coefficients of 0.97 and 0.95 respectively) as well as the extremes, considering the 95th percentile of maximum temperature

10  (Figure 7b) with a Pearson correlation of 0.95 and the 5th percentile of minimum temperature (Figure 7d) with 0.96.

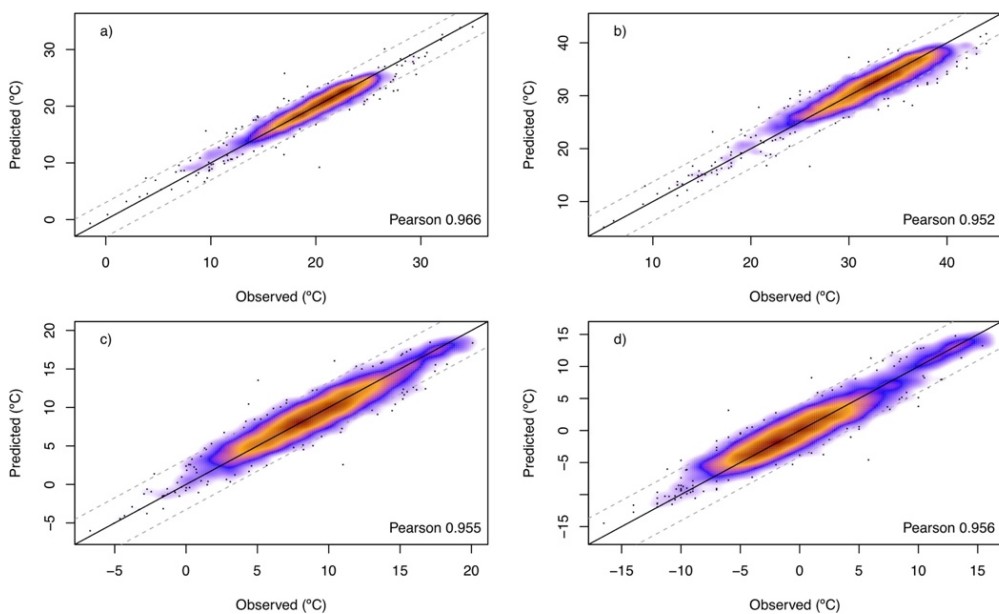

**Figure 7. Comparison between observations and estimates, by stations (n = 5520), of the mean maximum temperatures (a) and their 95th percentiles (b) and of the mean minimum temperatures (c) and their 5th percentiles comparison (d).**

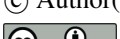


The mean Pearson correlations between the daily observations and the estimates, by months, were 0.87 and 0.82 in maximum and minimum temperature, respectively (Figure 8). However, more than 80% of the months in maximum temperature and more than 68% in minimum got a correlation higher than 0.8. Low correlations (Pearson < 0.5) represented a 3% and a 5% of the months in maximum and minimum temperature, respectively.

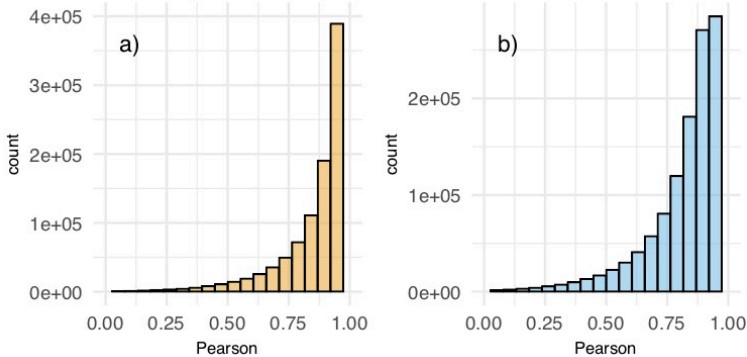

**Figure 8. Correlation frequencies between daily observations and estimates, by months, in the final dataset. Maximum (orange) and minimum (blue) temperatures are shown.**

The frequency of observed temperature and their estimates (Figure 9) showed a good general agreement. Although maximum
10  temperature was slightly overestimated in lower values (from 0 to 10 ºC), it was slightly underestimated in higher ones (from 20 to 35 ºC). The higher differences in minimum temperature were found in low values (an overestimation from -5 to 0 ºC) and in mid values (an underestimation from 10 to 20 ºC).

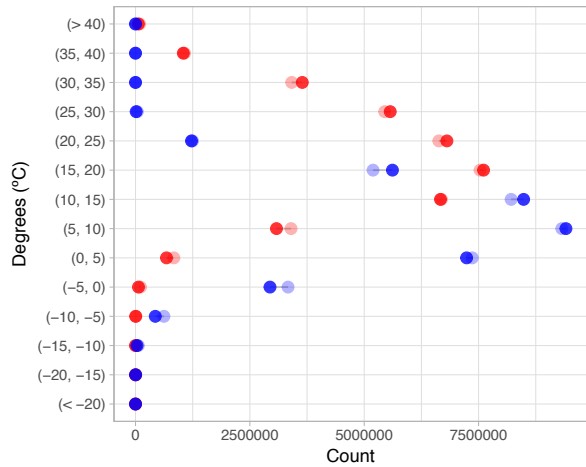

**Figure 9. Comparison of frequencies by categories between observed (solid) and predicted (transparent) maximum (red) and**
15  **minimum (blue) temperatures.**



In regard of monthly aggregates (Table 2), the ratio of means (RM) was 1 in all cases in TMAX, showing a similarity in the means between observed and estimated temperature, while the coldest months (November to February) showed a slight underestimation in TMIN. A bias in variance estimation was observed with ratios of standard deviation (RSD) under 0.95 in all months in TMAX and under 0.93 in TMIN. However, ME values were very low in both cases and ME values near to zero in all cases. Pearson correlations between observations and estimates were over 0.90 in all months in TMAX and ranging from 0.85 in July and August to 0.93 from November to February in TMIN.

**Table 2. The leave-one-out, cross-validation (LOO-CV) statistics showing the goodness of fit between maximum and minimum temperature observations and estimates of monthly aggregates. MAE: mean absolute error; ME: mean error; RM: ratio of means; RSD: ratio of standard deviations; Range: minimum and maximum observed temperature in each case. Results were constrained to 2 decimal places.**

|       |         | Jan   | Feb   | Mar   | Apr   | May   | Jun   | Jul   | Aug   | Sep   | Oct   | Nov   | Dec   |
|-------|---------|-------|-------|-------|-------|-------|-------|-------|-------|-------|-------|-------|-------|
| TMAX  | MAE     | 0.62  | 0.61  | 0.63  | 0.66  | 0.70  | 0.78  | 0.84  | 0.80  | 0.71  | 0.64  | 0.60  | 0.62  |
|       | ME      | -0.01 | -0.01 | -0.01 | -0.01 | -0.03 | -0.04 | -0.03 | -0.02 | -0.02 | -0.01 | -0.01 | 0.00  |
|       | RM      | 1.00  | 1.00  | 1.00  | 1.00  | 1.00  | 1.00  | 1.00  | 1.00  | 1.00  | 1.00  | 1.00  | 1.00  |
|       | RSD     | 0.88  | 0.92  | 0.92  | 0.92  | 0.93  | 0.92  | 0.89  | 0.88  | 0.91  | 0.94  | 0.91  | 0.87  |
|       | Pearson | 0.93  | 0.96  | 0.96  | 0.95  | 0.96  | 0.95  | 0.91  | 0.91  | 0.94  | 0.96  | 0.94  | 0.91  |
| TMIN  | MAE     | 0.81  | 0.81  | 0.82  | 0.77  | 0.79  | 0.83  | 0.90  | 0.90  | 0.87  | 0.84  | 0.81  | 0.81  |
|       | ME      | 0.01  | 0.02  | 0.01  | 0.03  | 0.03  | 0.05  | 0.02  | 0.02  | 0.03  | 0.01  | 0.01  | 0.02  |
|       | RM      | 0.94  | 0.96  | 1.00  | 1.00  | 1.00  | 1.00  | 1.00  | 1.00  | 1.00  | 1.00  | 0.99  | 0.96  |
|       | RSD     | 0.90  | 0.92  | 0.86  | 0.87  | 0.89  | 0.87  | 0.84  | 0.84  | 0.87  | 0.89  | 0.92  | 0.91  |
|       | Pearson | 0.93  | 0.93  | 0.88  | 0.89  | 0.89  | 0.88  | 0.85  | 0.85  | 0.88  | 0.91  | 0.93  | 0.93  |

The number of stations decreases as the elevation increases (Table 3). In Spain, only a 1.8% of temperature stations are over 2,000 m a.s.l. while a 37.40% are below 300 m a.s.l. This great difference, also shown in precipitation (Serrano-Notivoli et al., 2017a), necessarily affects the estimation of the variable. A slightly underestimation was observed in TMIN from 1,300 to 2,000 m a.s.l. and an overestimation in TMAX from 1,500 m a.s.l. The figures showed also a good agreement at all elevation ranges with the largest differences at high elevations (slight overestimation in TMAX and underestimation in TMIN). The MAE values were increased along with the elevation in TMAX from 0.65 to 1.21, while in TMIN were more constant. The ME also experimented an increase with the elevation in TMAX, but in TMIN all the values were near to zero.

**Table 3. The leave-one-out cross-validation (LOO-CV) statistics showing the goodness of fit between observations and estimates of daily maximum and minimum temperature separated by altitudes (m a.s.l.). N: number of stations; OBSm: mean observed temperature; PREDm: mean predicted temperature; MAE: mean absolute error; ME: mean error; RM: ratio of means; RSD: ratio of standard deviations. Results were constrained to 2 decimal places.**

|       |       | 0-100 | 100-300 | 300-500 | 500-700 | 700-900 | 900-1,100 | 1,100-1,300 | 1,300-1,500 | 1,500-2,000 | >2,000 |
|-------|-------|-------|---------|---------|---------|---------|-----------|-------------|-------------|-------------|--------|
| TMAX  | N     | 1,028 | 1,036   | 943     | 874     | 807     | 422       | 227         | 81          | 73          | 27     |
|       | OBSm  | 22.40 | 21.80   | 20.90   | 20.30   | 18.50   | 17.20     | 16.00       | 15.10       | 11.60       | 9.30   |
|       | PREDm | 22.70 | 21.90   | 20.90   | 20.40   | 18.50   | 17.20     | 16.00       | 15.20       | 11.90       | 10.10  |
|       | MAE   | 0.66  | 0.65    | 0.68    | 0.71    | 0.76    | 0.80      | 0.86        | 1.06        | 0.99        | 1.21   |





|  |  |  |  |  |  |  |  |  |  |  |  |
|---|---|---|---|---|---|---|---|---|---|---|---|
|  | ME | 0.07 | -0.03 | -0.06 | -0.04 | 0.01 | -0.07 | -0.09 | 0.21 | 0.36 | 0.54 |
|  | RM | 1.00 | 1.00 | 1.00 | 1.00 | 1.00 | 1.00 | 0.99 | 1.01 | 1.03 | 1.05 |
|  | RSD | 1.01 | 0.99 | 0.99 | 0.99 | 0.99 | 0.99 | 0.99 | 0.99 | 1.01 | 1.07 |
| TMIN | N | 1,028 | 1,036 | 943 | 874 | 807 | 422 | 227 | 81 | 73 | 27 |
|  | OBSm | 11.90 | 10.00 | 8.90 | 8.00 | 6.30 | 5.40 | 4.60 | 4.20 | 3.20 | 1.70 |
|  | PREDm | 12.10 | 10.10 | 8.90 | 8.00 | 6.30 | 5.30 | 4.70 | 4.10 | 2.30 | 1.40 |
|  | MAE | 0.82 | 0.80 | 0.82 | 0.86 | 0.85 | 0.95 | 0.97 | 0.97 | 0.91 | 0.88 |
|  | ME | 0.00 | 0.08 | 0.02 | -0.04 | 0.03 | 0.11 | -0.12 | -0.05 | -0.55 | -0.13 |
|  | RM | 1.00 | 1.01 | 1.00 | 1.00 | 1.01 | 1.02 | 0.97 | 0.99 | 0.88 | 0.92 |
|  | RSD | 0.99 | 0.99 | 0.99 | 0.99 | 0.99 | 0.99 | 0.99 | 0.98 | 0.95 | 0.96 |

Approximately, a 70% of the differences between observations and estimates both in TMAX and TMIN were lower than 1 ºC (Figure 10a), which assures the feasibility of the predicted series. Also, most of the spatial and temporal anomalies (approx. 80%) were lower than 1 (Figure 10b and c).

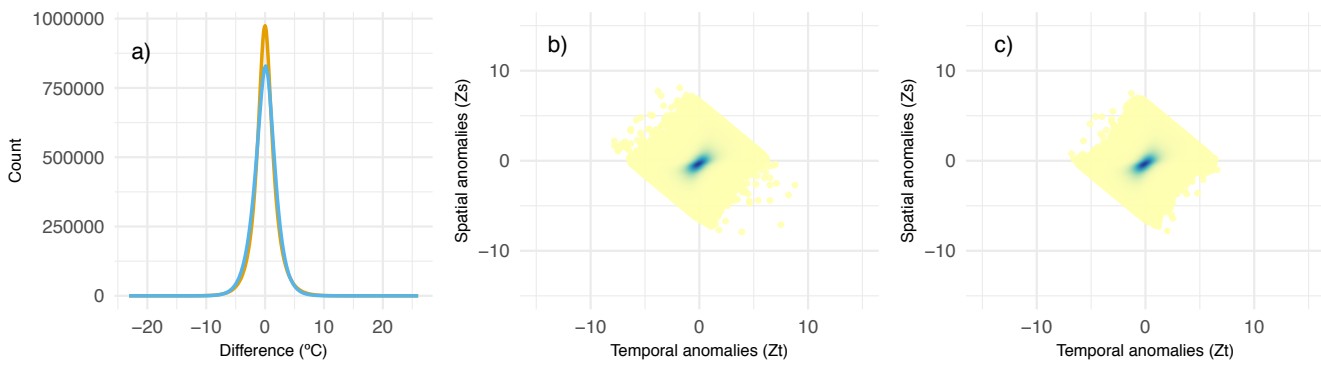

**Figure 10. Comparison of observations and estimates in the final dataset. Left column: maximum (orange line) and minimum (blue line) temperature differences between observations and estimates; central and right columns: temporal anomalies (Zs) vs spatial anomalies (Zs) of final dataset.**

10    The uncertainty of the estimates showed a decreasing temporal evolution (Figure 11a), with higher values at the beginning of the period coinciding with the moment of less observations and higher distance between them (see Figure 2a). The values in maximum temperature were lower than minimum until the 1950s an then they were similar during a couple decades until they converge at the end of 1980s, when they diverged and the uncertainty in maximum temperature decreased at a higher rate than minimum . Likewise, the annual mean error (Figure 11b) showed a great variability until the end of 1940s with similar values

15    in TMAX and TMIN, when they separate being TMAX over TMIN in the rest of the series. Since the 1970s, an approach between the two series is shown, being nearer to zero TMAX than TMIN, which is always in negative values.





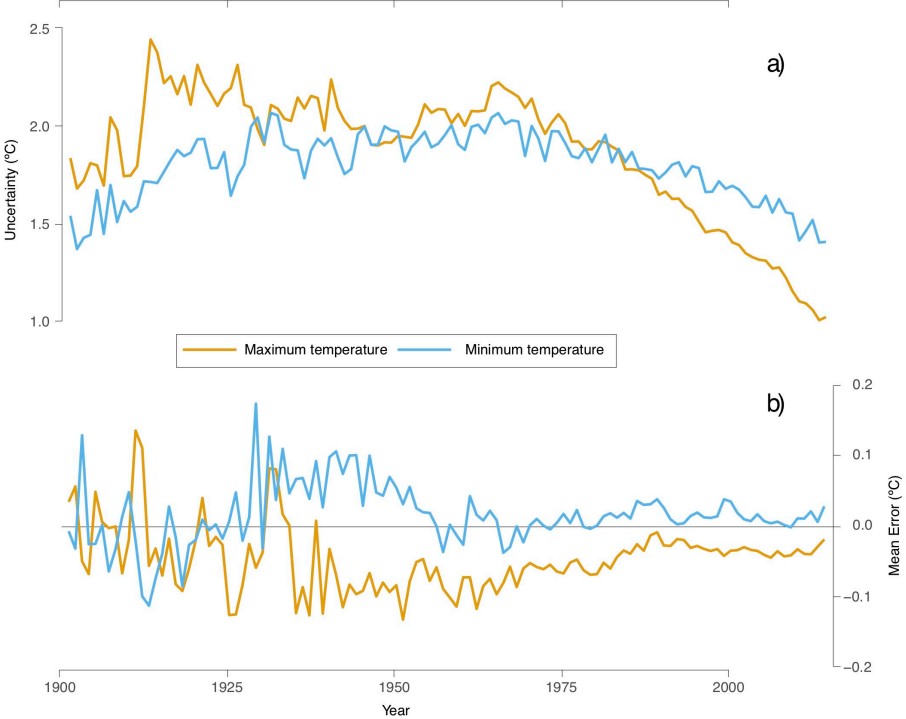

**Figure 11. Annual evolution of median daily uncertainty (a) and mean error (b) of maximum and minimum temperature in peninsular Spain.**

## 4.3 Spatial distribution and uncertainty of daily maximum and minimum temperature

The mean annual absolute daily maximum temperature (Figure 12a) showed a great variability, with highest values (> 44 ºC) in the central part of the Guadalquivir Valley and widespread areas with values over 40 ºC in southern half of Iberian Peninsula (IP), the lowest areas of the Ebro Valley and inner areas of larger islands, reflecting a continentality effect. The lowest values were found in highest elevations as the Pyrenees and the Iberian Range and also at northern IP. In this case, the uncertainty was inverse to the spatial distribution of the variable (Figure 12b), with higher values at north and at highest elevations in the Canary Islands, and lower in areas where maximum temperature is higher. On the other hand, the mean annual absolute daily minimum temperature presented a completely different spatial distribution (Figure 12c) especially in southern part of the IP, where there was a southwest-northeast gradient interrupted by high elevations of Sierra Nevada and Cazorla. The northern half of the IP showed a similar pattern than the previous index with coldest temperatures (< -10 ºC) coinciding with lower values of maximum temperature. The uncertainty of this variable (Figure 12d) was lower than the previous one with almost all the Spanish territory below 1 ºC. The lowest values were found at northwest IP and the higher ones in coastal areas of Mallorca.



### 4.4 Spatial distribution and uncertainty of frost days and tropical nights

The mean annual number of frost days (Figure 12e) varied from less than 10 in coastal areas of IP and in all Balearic and Canary Islands, to more than 200 in highest elevations of the Pyrenees. Between these extremes, a similar increasing gradient as the minimum temperature was found in the southern part of the IP and in the Ebro Valley, while the northern plateau was

dominated by a range of 50-100 days. The spatial distribution of the uncertainty (Figure 12f) coincided with the variable, with highest values where the number of frost days was higher. However, some exceptions were found: one at northeast IP, with a high uncertainty of relatively low number of frost days; and other at high elevations of the Pyrenees, where the uncertainty was low in regard of the high number of days.

The mean annual number of summer days (Figure 12g) showed a similar spatial pattern than the maximum temperature but

with a stronger effect of the orography. The highest values were found in the Guadalquivir Valley with more than 150 days, as well as in southern part of the Mediterranean coast and eastern Canary Islands. The lowest number of summer days (< 25) coincided with highest elevations of Central and Iberian Range, Pyrenees and Sierra Nevada. Also, all along the Cantabric coast showed values lower than 75 summer days. The uncertainty related to this index (Figure 12h) was higher than the frost days, with a clear gradient from less than 2 days in central southern IP to more than 3 days in all northern IP, the Iberian Range,

the Canary Islands and most of the inner areas of the Balearic Islands.









**Figure 12. Mean annual (1981-2010) values for a) absolute daily maximum temperature; c) absolute daily minimum temperature; e) number of frost days and; g) number of summer days and their corresponding uncertainty (b, d, f and h, respectively).**

## 5 Discussion

This work introduces two important novelties in regard of high-resolution climatic analysis: i) a new methodology to reconstruct *in situ* temperature data series over time and space, and ii) a new daily gridded temperature dataset for Spain. The method, which is based on reference values (RV) computed from nearest observations instead of reference series (RS), follows the protocol developed for precipitation reconstruction in Serrano-Notivoli et al. (2017b). However, we included the distance to the coast as a source of variation of the local models in addition to the three used in the previous work (altitude,

latitude and longitude). This parameter has been proved to be important for temperature estimation (Fick and Hijmans, 2017) and lets the models be more flexible.

One of the most valuable keys of the approach presented here is the use of all the available climatic information, which is crucial for a high-resolution output due to the observations network density has a major influence in gridded datasets results, controlling the skill of the final estimate of the variable (Hofstra et al., 2008). This is especially true in high percentiles, with

a disproportionate effect in extreme values and, therefore, in extreme indices (Hofstra et al., 2010). Hence, a method using all the information instead of longest data series seems appropriate. Indeed, there are several temperature estimation methods in literature, and the choice of one or another is not a trivial matter since the gridded dataset will be built from estimates. The inference or interpolation of any climatic variable in different locations from the recording sites always implies some kind of variation in final estimates regarding the observations. The aim is, therefore, using an approach minimizing these errors.

Previous comparatives of interpolation methods do not conclude on any definitive one. For instance, Shen et al. (2001) make a review of daily interpolation methods resolving that almost all of them smooth the data, and Jarvis et al. (2001) did not found large differences between them either. However, Hofstra et al. (2008) accept as more appropriate a global kriging for they work at European scale and others as Jeffrey et al. (2001) use simpler methods as thin splines.

In this work, we use GLMMs and GLMs as a general approach to the daily temperature estimation, using as support monthly

estimates based on daily data of months with complete observations. This part gives consistency to all the temporal structure of the data series, as similar approaches used in previous works (e.g. Jones et al., 2010). On the other hand, the use of regressions in temperature estimation is not new. For example, several works establish that regression models are more reliable than other interpolation methods for monthly temperatures (Kurtzmann and Kadmon, 1999; Güler and Kara, 2014). Li et al. (2018) built a high-resolution grid for urban areas in USA using geographically weighted regressions (GWR) and reported

Pearson correlations between 0.95 and 0.97, similarly to the present work. However, Hofstra et al. (2008) found that, for European scale, daily temperature regressions worked worse than other interpolation methods.

The present work constitutes a novelty regarding previous methodological approaches mainly due to: i) all the available information is used, being the longer series supported by shorter ones, and ii) it includes a comprehensive iterative quality



control checking the spatial and temporal consistency of the data until no suspicious values are detected. In addition to the developed validation process, the results in the form of spatial coherence show that the method is able to reproduce realistic climatic situations. The new approach of the quality control detects a number of suspect data in line with previous research, assuring the deletion of anomalies in a spatial and a temporal dimension. Although many works dedicate little efforts to this

part of the reconstruction, it is one of the most important since it will have a decisive weight in the final result. For instance, Jeffrey et al. (2001) simply remove those data exceeding a fixed threshold in regard of the residuals of the splines; and in ECA&D (Klok and Klein-Tank, 2009) the quality tests are absolute, without a comparison with neighbouring data series. Nevertheless, a similar approach to our spatial check of quality was developed in Estévez et al. (2018) but comparing nearest stations in all their data series instead of daily individual data. They used the spatial regression test (SRT) following You et al

(2008) and Hubbard et al. (2005). Others such Durre et al. (2010) also applied spatial consistency checks to test if temperature data lies significantly outside their neighbours. They flagged as suspect a 0.24% of all the (worldwide) data considering temperature, precipitation, snowfall, and snow depth, a quite low figure.

One of the key elements in any gridded dataset creation is to provide an uncertainty value, which informs about the reliability of the data and should be a standard to all the climatic information. The uncertainty values presented in this work comes from

each of the individual models for each timestep and location, therefore it apprises about the changes in the reliability of the day-to-day data. Now, several datasets provide this kind of information, though it can be obtained from different methods. For example, Cornes et al. (2018) applied a smart calculation of the uncertainty using 100-member ensemble realizations for each day; Stoklosa et al. (2015) and Di Luzio et al. (2008) used PRISM (Parameter–Elevation Regressions on Independent Slopes Model) to compute uncertainty in two ways: i) a leave-one-out cross-validation (LOO-CV) as we do in the present work, and

ii) modeling the uncertainty –which could lead to a propagation of the errors– using the prediction intervals of their weighted linear regression. In all cases the method is valid because the goal is to extract the potential bias for each considered timestep. Concerning to the new dataset, although some previous works created daily temperature datasets for Spain, only a few are gridded (only Herrera et al., 2016 built one for whole peninsular Spain and Balearic Islands) and none of them are dedicated to analyse the spatial distribution of daily temperature indices but the trends (e.g.: El-Kenawy et al., 2011; Fonseca et al.,

2016). We show here only four examples of the capabilities of STEAD dataset in the research of temperatures in Spain. The northern half of the IP showed a stronger influence of orography and Atlantic influences, just like in annual precipitation and maximum precipitation in 1 and 5 days (SPREAD, Serrano-Notivoli et al., 2017a), showing a potential covariability with other precipitation indices or temporal scales (Sánchez-Rodrigo, 2018 and 2014; Fernández-Montes et al., 2016). Besides, the availability of maximum and minimum temperature (STEAD) and precipitation (SPREAD) at same temporal (daily) and

spatial (5x5 km) scale, opens up possibilities of new prospective research in many fields as agricultural climatology, natural hazards, paleoclimatic reconstructions or hydrological modelling, amongst others.

In our attempt to create a useful reconstruction and gridding methodology, some of the stages of the method imply arbitrary decisions that could be changed based on user-defined options. For instance, we use 15 neighbouring observations to build the model but there is not an objective number. We are building these models with 4 cofactors, which need certain degrees of

freedom. An increase in the neighbours could lead to a loss of local representativeness, but also a gain of statistical robustness and lower influence of anomalous data.

In respect of the quality control process, the initial thresholds were set at the beginning only to remove outliers that are sometimes included in the original datasets (e.g. -999 or nonsenses as 54354 that is one of the removed values in this work)
but that sometimes have a meaning to identify specific situations or local codes. There are a lot of sources and types of errors as repeated series, duplicities or coding errors, that we try to identify through a simple collection of criteria. For example, we use the correlation between series and the differences between observations and estimates to remove data based on probability thresholds that we defined based on our experience, but maybe others could be useful depending on the dataset.

The effort of this research has been mainly dedicated to create an accurate estimate of temperature using all the available
information and providing a validation as complete and transparent as possible, as well as afford an uncertainty measure tailored for each value allowing the assessment of data in each day and location.

## 6 Data availability

The STEAD dataset is freely available in the web repository of the Spanish National Research Council (CSIC). It can be accessed through http://dx.doi.org/10.20350/digitalCSIC/8622, and cited as Serrano-Notivoli et al. (2019). The data is arranged
in 12 files (daily maximum and minimum temperature estimations and their uncertainties for peninsular Spain, Balearic Islands and Canary Islands) in NetCDF format that allows an easy processing in scientific analysis software (e.g. R, Python…) and GIS (list of compatible software at http://www.unidata.ucar.edu/software).

## 7 Conclusions

We present a new high-resolution daily maximum and minimum temperature dataset for Spain (STEAD). Using all the
available daily temperature data, a 5 x 5 km spatial resolution grid was created. The original data were quality-controlled and the missing values were filled based on the monthly estimates and using the 15 nearest observations. A serially complete dataset was obtained for all stations from 1901 to 2014 for peninsular Spain and from 1971 to 2014 for Balearic and Canary Islands. Based on this dataset, daily temperatures were calculated for each grid node, resulting in a high-resolution gridded dataset that we used to compute four daily temperature indices: mean annual absolute maximum and minimum temperatures,
mean annual number of frost days and mean annual number of summer days.

The spatial distribution of mean annual maximum and minimum temperatures showed a strong relationship with the altitude, (decreasing along with the elevation) and with the distance to the coast, revealing a high effect of continentality with increased values of the indices in both inner mainland Spain and islands. The mean annual number of frost days was higher in northern half of peninsular Spain and in high-elevation areas of the south, while the mean number of summer days obtained the highest
values at south, in the Guadalquivir Valley and southern Mediterranean coast, progressively decreasing to the north.



The use of all the available information in combination with a methodology based on local variations of temperature over a high-resolution grid, provided a daily dataset that is able to reproduce the high spatial and temporal variability.

**Author contribution**

M. de Luis designed the methodological approach in collaboration with R. Serrano-Notivoli, who applied it to the reconstruction of the climate data and the development of the gridded dataset. S. Beguería contributed to the validation process and the climatic analysis of the results. R. Serrano-Notivoli prepared the manuscript with contributions from all co-authors.

**Acknowledgements**

This study was supported by research projects CGL2015-69985-R and CGL2017-83866-C3-3-R, financed by the Spanish Ministerio de Economía y Competitividad (MINECO) and EU ERDF funds. R.S.N. is funded by postdoctoral grant FJCI-2017-31595 of the 'Juan de la Cierva' Programme, funded by the Spanish Ministry of Science, Innovation and Universities, and EU ERDF. The authors thank the Spanish Meteorological Agency (AEMET) for the data.

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
