# Peer review of "STEAD: A high-resolution daily gridded temperature dataset for Spain"

_Earth System Science Data, 2019_

## Short Comment (SC1)

**RESPONSE LETTER**

Review of Manuscript No.: essd-2019-52
Title: STEAD: A high-resolution daily gridded temperature dataset for Spain
Author(s): Serrano-Notivoli R, Beguería S, de Luis M

**GENERAL COMMENTS**

**Overall this is a thorough piece of research, and the content is entirely suitable for this journal. The gridding method, including the quality control procedure, are well documented, but some clarifications are necessary prior to publication.**

Thank you for your comments. We tried to be clear in the description of the methodological process and the resulting products. We hope that the changes made in the manuscript are fair enough for a straightforward understanding.

**SPECIFIC COMMENTS**

**A common problem with datasets that grid tmax and tmin separately is that there is no guarantee that tmax will be greater than tmin in the final dataset. I have checked the dataset and there are several days where tmax<tmin values occur in certain grid cells. These mostly occur across the edges of the gridding domain. The highest frequency (11) of tmax<tmin is during the year 1996, although such occurrences are apparent for most years. I do not advocate changing the methodology to account for this, but this limitation needs to be highlighted in the paper.**

Thank you for checking carefully the dataset, we appreciate the concern. We realized that the available dataset at the *digitalCSIC* repository is a previous version rather than the final one. We have uploaded the correct version, which was used to make the analysis in the manuscript. Sorry for the mistake. The methodological process creates new estimates for maximum and minimum separately. However, all the stages, from the quality control to the gridding, consider the differences between them to regularly check the internal consistency, so it is impossible to get situations in which tmax<tmin.

**The methodology is broadly similar to the method used in the SPREAD precipitation dataset, produced by the same authors. I refer to the use of Reference Values and Generalized Linear Models, as for precipitation the skewed nature of the data and zero-cutoff were also taken into account. Nonetheless, given that there is a connection between STEAD and SPREAD I think that the precipitation dataset needs to be mentioned earlier in the introduction, and in the Methods Section the differences in the method used here for the temperature variables should be indicated.**

Based on your suggestion, we added an extended explanation of SPREAD-STEAD relationship in the introduction:

*"The experience acquired in the SPREAD dataset (Serrano-Notivoli et al., 2017a) development set the basis for a solid and reliable daily gridded precipitation datasets creation. Using the same framework with a complete renewal of the core calculations, we developed a new methodology for daily temperature datasets reconstruction and grids creation."*

and also at the beginning of the methods section:

*"The key stages of the methodological process (calculation of RV, quality control, gap filling and gridding) are the same to that used to create the SPREAD dataset (Serrano-Notivoli et al., 2017a). However, the method basics are completely different since the RV creation has been refined, the quality control has been adapted to temperature data, and*

*the gap filling and gridding processes include now an improved standardization procedure."*

**Section 2: Information needs to be provided about the time schedule over which the daily maximum and minimum values were recorded. This may not be available for all stations but where available it should be described briefly in this section, e.g. are tmax/tmin calculated over the full 24-hour period and does this change over time.**

Since we used the daily products provided by AEMET and MAGRAMA, we don't have the information about the moment of the day in which maximum and minimum temperatures were recorded. Anyway, this issue doesn't affect to the temperature estimates because the method creates a prediction for each original observation, independently of when it was recorded. The final reconstructed series faithfully represents the temporal structure of the original ones, regardless of the moment of recording.

We added a sentence in section 2 explaining that we used the daily maximum and minimum values of temperature observations:

*"Daily maximum and minimum values of temperatures series were used from all the observatories."*

**Section 2: As pointed out by reviewer #2, the changing number of input stations over time can have a profound influence on the gridded data, and is important for users who want to calculate long-term trends from the data to be aware of this. This limitation of the dataset needs to be highlighted.**

We contributed to the minimization of the impact of the data availability over the time introducing a standardization procedure. The methodological approach creates spatial references that are standardized with the temporal structure of the series to avoid biases or incoherencies. Furthermore, the provided uncertainty values for each of the estimates inform about the reliability of the data. This was already mentioned in the discussion section (6th paragraph). However, we emphasized this interesting subject in section 2:

*"[…] Despite the differences in the data availability through time, the methodological process creates spatial references that are standardized with the temporal structure of the series to avoid biases or incoherencies. In this regard, the chosen spatial resolution accurately reflects the local characteristics of daily temperature in most of the temporal period, while the provided uncertainty values help to understand the reliability of the estimates when the original data have higher variability."*

**Section 4.2: The verb "depurate" appears to me to be wrongly used for this procedure. Suggest changing to simply "Quality-controlled dataset".**

Modified as suggested.

**Section 5 (Discussion) lines 20-23: The key point about producing these gridded datasets is that the final values should reflect grid-box average values that are based on limited spatial sampling (unless the method produces values representative of point- values, which I understand that it does not). This relates to the comments by reviewer #2 about the choice of gridding resolution. This general aim of gridding is not articulated well in this discussions section and needs revision.**

Whereas your comment is very interesting, maybe there is a misunderstanding at this point. The temperature estimates (as well as their corresponding uncertainty values) are created for specific individual locations represented by 4 parameters (i.e. latitude, longitude, altitude and distance to the coast). The representativeness of the grid-box in each case falls on that each of those parameters are the median value of all covering that area. For instance, in the STEAD dataset, each gridpoint is the centroid of a squared area of 5 x 5 km with the median of all the possible values of the parameters covering that area.

We added an explanation in the description of the grid (section 2) to avoid misunderstandings:

*"[…] The predictor parameters (i.e. latitude, longitude, altitude and distance to the coast) for each grid point were computed as the median of all the possible values of those parameters, covering an area of 5 x 5 squared km in which the grid point is the centroid. […]"*

**Abstract and Conclusions: One of the key aspects of this paper is the use of many more stations than used in other datasets for the region. This needs to be stated more clearly in the abstract and conclusions, as the phrase "full total of available 5520 observatories" does not convey to the reader (especially those not familiar with the station density across the region) that this is an important feature of this dataset.**

Despite the full total available number of stations is 5,520, this figure is for the whole period, so the station density has greatly varied though time. However, we included in the abstract and conclusion, as suggested, the theoretical density:

*"[…] (about 1 station per 90 km2 considering the whole period) […]"*

---

## Referee Comment (RC1) · Anonymous Referee #1 · 3 May 2019

The manuscript entitled "STEAD: A high-resolution daily gridded temperature dataset for Spain" shows a very serious analysis of daily precipitation spatial and temporal behaviour in an area where rainfall has been widely studied.

I really enjoyed reading this work about the methodological procedure for generating a high resolution gridded dataset for temperatures in Spain. The paper is, overall, very good and very well written. The objectives proposed are carried out rigorously following a proper structure. Moreover, the discussion is very well developed and carries out a very interesting deepening about the implications of considering different criteria to develop the dataset. The results shown are very consistent.

I do really appreciate the exhaustive quality control over daily temperature data base on paired comparisons between observations and standardized predictions shown in

section 3.3, as well as the consideration of the distance to the coast as a source of variation of the local models.

I think that it fits the scope of the journal and can be published as it is, only with very with few corrections.

- p. 2 l. 24: "leads to high risks related to..." risks such as? - p. 5 l. 5-6: can you explain why you chose these thresholds or it is a subjective criteria? - p. 10 l. 2: "and" instead of "&", the double condition is more understandable this way - p. 13: I suggest to move Table 1 to the previous page

---

## Referee Comment (RC2) · Anonymous Referee #2 · 4 May 2019

General Comments

This manuscript describes a gridded temperature data set at daily and 5 km x 5 km spatial resolution for Spain. Compared to existing similar data sets it represents a valuable contribution considering the data set temporal and spatial resolution, the period covered (1901 to 2014 for continental Spain) and the powerful methodology employed, which favours the use of a high number of stations and the computation of uncertainty of the estimated variables. Overall I think it is a well written manuscript and the data set may indeed be useful for future studies, therefore it is a perfectly suited article and data set for this journal.

However, I think it requires some clarifications and corrections before being acceptable for publication. For example it is not clear to this reviewer why the spatial resolution

used was chosen and what is the impact of the variable station number (and therefore density) upon the quality of the estimates during the different years considered in the dataset. These details may influence further studies (such as in the interpretation of temporal trends) so I think they should be commented here, in the description of the data set proposed. As explained in more detail below, I propose to modify or add some of the existing figures related to this aspect and also to the global uncertainty and mean error time series.

Another aspect that should be clarified are the data sources. The manuscript describes two institutions (AEMET and MAGRAMA) but in the data set description link provided by the authors in the manuscript, if I understood correctly, 12 providers are listed (which, according to descriptor "dc.relation.isbasedon" are: Spanish Meteorological Agency (AEMET); Ministry of Agriculture and Environment; Servei Meteorològic de Catalunya (METEOCAT); Navarra Government; SAIH Cantábrico; SAIH Duero; SAIH Ebro; SAIH Guadalquivir; SAIH Hidrosur; SAIH Júcar; SAIH Miño-Sil; SAIH Segura; SAIH Tajo), so that implies 10 sources more than those mentioned originally in the manuscript. Obviously one of the two descriptions is not correct so this should be clarified and corrected.

Finally, there are a number of formal aspects that should be amended such as the use of correct symbol units, references quoted in the text but not listed in the references section, or minor problems with English language (I do not intend to be exhaustive in this aspect but I list some issues below). For all the above, despite this is a very interesting contribution, I do not think the manuscript can be accepted in its current form and I recommend major revision.

Specific Comments

1. Page 2, line 20. Please add here the horizontal resolution of the E-OBS dataset, i.e. "with a horizontal resolution of .." or something similar.

2. Page 2, lines 23. The first part and last parts of the sentence "The Spanish territory

perfectly captures... this variability" is unclear and potentially misleading... what do you mean by "the great climatic variability"? The climatic variability of the world? I suggest to rewrite it, for example "The Spanish territory exhibits a great climatic variability with very different regimes in a relatively small area that leads to high risks such as.." (please complete with what you think are those risks) or something similar.

3. Page 3, line 13. Please complete the sentence with all data sources (as mentioned earlier) if there are more than two - or correct the 12 data sources given in the data set description repository.

4. Page 3, figure 1. Map bottom panel of Canary Islands: please enlarge axis font sizes they can hardly be read

5. Page 4, line 2. Please give a range of mean distances between stations, as it is done later for mean station elevations.

6. Page 4, Figure 2. This is a very important part of the data set description, as it gives an idea of temporal changes in mean altitude, minimum distance and number of stations. The methodology used by the authors allows the use of a variable number of stations, not constant in time, which is good because it maximizes the information introduced in the estimation. However, this introduces variability due to the changes in the station data set so this should be carefully described. Authors chose to show mean values only, which I think it is rather limited, and combined in the bottom panel two variables. I suggest to consider a three panel figure, one for each of the three variables considered, and to plot for each one the median and percentiles 25 and 75 - alternatively, if variables examined follow a Gaussian distribution mean values plus standard deviation could be also a possibility - however I would favour the first option. Then the new figure should be briefly commented, and in particular, justify properly (here or in section 4.1) the selection of the 5 km x 5 km horizontal resolution, which in the current version seems arbitrary in this section (in the discussion the reader founds that it is consistent with an already existing precipitation data set for the same region,

but this is not either mentioned in section 2 and could be also considered).

7. Page 5, section 3.1. Please introduce properly the meaning of TMAX and TMIN.

8. Page 10, line 25. How many iterations of the procedure described are typically required?

9. Page 11, line 17. Pearson -> Pearson correlation coefficient ? [at least the first time you mention it]

10. Page 14, section 4.1. Please justify or just comment briefly (if justified previously) the selection of the horizontal resolution.

11. Page 14, line 7. Suggest: case -> case using a leave-one out cross validation (LOO-CV) [this is mentioned later but not in the text]

12. Page 14, line 9. I think there is a rounding problem here and 0.95 should be 0.96 according to figure 7c - please check.

13. Page 14, figure 7. It is good the authors chose to show the four panels with the same x-axis and y-axis ranges to compare them properly. However, the panels are shown as rectangles and not as squares thus using a different scale for x and y axis. Could you please fix this?

14. Page 14, figure 7. Please indicate (in the figure caption is fine) the meaning of the dashed lines - confidence intervals perhaps? In that case specify the level.

15. Table 2 (and 3). Please use correct units where needed.

16. Table 2 title. When enumerated the variables displayed on the table, the last one is "Range: minimum and maximum..." but on the last row, first column, it appears "Pearson". Please correct. Note that including the units, as requested above, may help detect these problems.

17. Table 2 (and 3). When you say decimal places you mean decimal digits? Please

check.

18. Table 3. Suggest adding in the first row, first column, the label "Altitude (m)", referred to the values listed on the first row, next columns.

19. Page 18, Figure 11. This is a very important part of the study and I think it deserves more attention. The uncertainty (upper panel) initially increases, ca. from 1900 to 1905, and then decreases from 1975. However, if I understood correctly, authors mention only a decreasing trend (page 17, line 10). Could you please clarify this? I suggest adding a background grid to the figure to allow an easier visual analysis.

20. Page 18, Figure 11. I think that, given the variability of terrain heights in Spanish territory and different station densities at different altitudes, it is necessary to stratify Figure 11 into an additional 6 panel figure, considering to split the aggregated uncertainty and mean error values into different station altitudes. Looking at Table 3 altitude classes, probably 3 station groups, for example those with altitudes contained in the following intervals [0, 500), [500, 1500), [1500, ) m above sea level, would be enough. Future studies examining aspects at different terrain heights may largely benefit from this additional figure to better interpret subsequent results.

21. Results given in sections 4.3 and 4.4 refer both to Figure 12. I suggest to split that figure into two figures (first and last four panels respectively) so that it is easier to read the comments referred to each part of the figure.

22. Regarding Figure 12f and 12h I noted that the uncertainty values (expressed in days) over the islands (both Balearic and Canary Islands) are either very low (Figure 12f) -except for the highest terrains in Canary islands - or very high (Figure 12h). Could you please comment this result?

Technical Comments & Minor Details

23. Page 1, line 24. Typo?: team -> teams? Please check meaning and correct if necessary.

24. Page 1, line 26. Jones et al 2010: reference quoted but not listed in references section. Is it perhaps Jones et al 2012?

25. Page 1, line 26. Please check citation journal style: Willmott and Matsura (1900-2014) (2001) -> Willmott and Matsura (2001) ?

26. Page 1, line 29. Check citation journal style: 2015 and 2018 -> 2015; 2018?

27. Page 2, line 13 and line 15. "e.g.:" -> "e.g." as in line 9, same page?

28. Page 2, line 27. English: did not considered -> did not consider

29. Page 3, line 12. check: down -> bottom map panel

30. Page 3, line 12. Aemet -> AEMET

31. Page 3, line 18. English, suggest: this moment -> then [or "that moment"]

32. Page 5, Figure 3 caption. Suggest: RV -> Reference Values (RV) [I know it is already defined in the text, but this change improves the readability of the figure]

33. Page 5, section 3.1. Please use correct Celsius degree symbols as you have done elsewhere in the manuscript.

34. Page 5, line 7 (and elsewhere in the manuscript). English: please check the meaning of suspected, suspect and suspicious and use properly.

35. Page 8, line 8. English: supplemental -> supplemental material

36. Page 9, line 1. English: finish -> finishes

37. Page 10. For some reason, in this page, text before equations end with a "." and not ":" as it is done elsewhere in the manuscript - please check.

38. Page 11, line 2. English, suggest check: obtain -> obtaining

39. Page 12, line 11. English: "fulfil" is correctly spelled, but it is in the British form - being the American form "fulfill". As you use previously in the manuscript "neighbor"

(American form) this is not consistent: you should chose either American or British forms, but not a mixture.

40. Page 13, line 5. English: what -> which? Please check and correct if necessary.

41. Page 13, Figure caption 6. upper (bottom) line -> upper (bottom) row

42. Page 16, line 14 (and elsewhere in text): a X% -> X% (remove "a" if only values are given)

43. Page 16, line 15. English: slightly -> slight [as in line 17 same page]

44. Page 17, line 2. Please expand "approx."

45. Page 18, line 13. Sentence "The northern half..", please check English - by "than" you mean "to"?, otherwise compared to what?

46. Page 18, last line. higher -> highest?

47. Page 21, line 14. Hofstra et al 2008: reference quoted but not listed in the references section. Idem in same page references Jarvis et al 2001, Jones et al 2010 (is it 2012?), Kurtzmann and Kadmon 1999 (is it 2009?) (next page) Hubbard et al You et al ... please check carefully and make necessary corrections.

48. Page 21, line 11. English: lies > lied?

49. Page 21, line 14 English: comes -> come

50. References: please check alphabetical order - the last one (Van Den ... ) should not be there.

---

## Referee Comment (RC3) · Anonymous Referee #3 · 15 May 2019

\* General Comments

Overall this is a thorough piece of research, and the content is entirely suitable for this journal. The gridding method, including the quality control procedure, are well documented, but some clarifications are necessary prior to publication.

\* Specific Comments

- A common problem with datasets that grid tmax and tmin separately is that there is no guarantee that tmax will be greater than tmin in the final dataset. I have checked the dataset and there are several days where tmax<tmin values occur in certain grid cells. These mostly occur across the edges of the gridding domain. The highest frequency (11) of tmax<tmin is during the year 1996, although such occurrences are apparent for

most years. I do not advocate changing the methodology to account for this, but this limitation needs to be highlighted in the paper.

- The methodology is broadly similar to the method used in the SPREAD precipitation dataset, produced by the same authors. I refer to the use of Reference Values and Generalized Linear Models, as for precipitation the skewed nature of the data and zero-cutoff were also taken into account. Nonetheless, given that there is a connection between STEAD and SPREAD I think that the precipitation dataset needs to be mentioned earlier in the introduction, and in the Methods Section the differences in the method used here for the temperature variables should be indicated.

- Section 2: Information needs to be provided about the time schedule over which the daily maximum and minimum values were recorded. This may not be available for all stations but where available it should be described briefly in this section, e.g. are tmax/tmin calculated over the full 24-hour period and does this change over time.

- Section 2: As pointed out by reviewer #2, the changing number of input stations over time can have a profound influence on the gridded data, and is important for users who want to calculate long-term trends from the data to be aware of this. This limitation of the dataset needs to be highlighted.

- Section 4.2: The verb "depurate" appears to me to be wrongly used for this procedure. Suggest changing to simply "Quality-controlled dataset".

- Section 5 (Discussion) lines 20-23: The key point about producing these gridded datasets is that the final values should reflect grid-box average values that are based on limited spatial sampling (unless the method produces values representative of point-values, which I understand that it does not). This relates to the comments by reviewer #2 about the choice of gridding resolution. This general aim of gridding is not articulated well in this discussions section and needs revision.

- Abstract and Conclusions: One of the key aspects of this paper is the use of many

more stations than used in other datasets for the region. This needs to be stated more clearly in the abstract and conclusions, as the phrase "full total of available 5520 observatories" does not convey to the reader (especially those not familiar with the station density across the region) that this is an important feature of this dataset.

---

## Author Response (AR1)

**RESPONSE LETTER**

Review of Manuscript No.: essd-2019-52
Title: STEAD: A high-resolution daily gridded temperature dataset for Spain
Author(s): Serrano-Notivoli R, Beguería S, de Luis M

**GENERAL COMMENTS**

The manuscript entitled "STEAD: A high-resolution daily gridded temperature dataset for Spain" shows a very serious analysis of daily precipitation spatial and temporal behaviour in an area where rainfall has been widely studied.

I really enjoyed reading this work about the methodological procedure for generating a high resolution gridded dataset for temperatures in Spain. The paper is, overall, very good and very well written. The objectives proposed are carried out rigorously following a proper structure. Moreover, the discussion is very well developed and carries out a very interesting deepening about the implications of considering different criteria to develop the dataset. The results shown are very consistent.

I do really appreciate the exhaustive quality control over daily temperature data base on paired comparisons between observations and standardized predictions shown in section 3.3, as well as the consideration of the distance to the coast as a source of variation of the local models.

I think that it fits the scope of the journal and can be published as it is, only with very with few corrections.

> Thank you for your comments. We have been working in the development of the methodology for a long time, testing different approaches with different parameters. The present version is the most accurate given the great variety of temperature values in a large dataset as the Spanish network.

**SPECIFIC COMMENTS**

**- p. 2 l. 24: "leads to high risks related to..." risks such as?**

> We added a short sentence in this part of the text to clarify the type of risks.

> *"[…] such as increments in the frequency and magnitude of extreme events."*

**- p. 5 l. 5-6: can you explain why you chose these thresholds or it is a subjective criteria?**

> The thresholds are based on the maximum and minimum absolute records of the Spanish Meteorological Agency (AEMET). This organism checks the reliability of the absolute temperature extremes, being the official ones -30.0 ºC (1963) for minimum temperature and +46.7 ºC (2017) for maximum temperature. We used -35 and +50, respectively, to discard wrong data.

**- p. 10 l. 2: "and" instead of "&", the double condition is more understandable this way**

> Modified as suggested.

**- p. 13: I suggest to move Table 1 to the previous page**

> Thank you. We agree that the table should be near to the beginning of section 4.1. We will report this back to the responsible of the publication production process.

**RESPONSE LETTER**

**Review of Manuscript No.: essd-2019-52**
**Title: STEAD: A high-resolution daily gridded temperature dataset for Spain**
**Author(s):** Serrano-Notivoli R, Beguería S, de Luis M

**GENERAL COMMENTS**

**This manuscript describes a gridded temperature data set at daily and 5 km x 5 km spatial resolution for Spain. Compared to existing similar data sets it represents a valuable contribution considering the data set temporal and spatial resolution, the period covered (1901 to 2014 for continental Spain) and the powerful methodology employed, which favours the use of a high number of stations and the computation of uncertainty of the estimated variables. Overall I think it is a well written manuscript and the data set may indeed be useful for future studies, therefore it is a perfectly suited article and data set for this journal.**

> Thank you very much for your nice and constructive comments. Our aim was to create a useful dataset for a broad community of researchers, not only climatologists. We are happy to receive such a complete feedback of our research that we are sure that it will greatly improve the final manuscript.

**However, I think it requires some clarifications and corrections before being acceptable for publication. For example, it is not clear to this reviewer why the spatial resolution used was chosen and what is the impact of the variable station number (and therefore density) upon the quality of the estimates during the different years considered in the dataset. These details may influence further studies (such as in the interpretation of temporal trends) so I think they should be commented here, in the description of the data set proposed. As explained in more detail below, I propose to modify or add some of the existing figures related to this aspect and also to the global uncertainty and mean error time series.**

> With the aim of make clear the choice of the 5 x 5 km spatial resolution of the gridded dataset, we added an explanation at the end of the "Input data" section. Furthermore, we also explain that, despite that the methodology includes a final stage when the estimates are standardized with the temporal structure of the original series, minimizing the impact of the number of stations. In addition, the final dataset provides an individual uncertainty value for each daily estimate, which informs about the reliability of the estimates through time. In regard of the use of data for temporal trends, both the standardization of the estimates based on the original series and the associated uncertainty values, provide the required information to assess the long-term trends of each series of estimates.

**Another aspect that should be clarified are the data sources. The manuscript describes two institutions (AEMET and MAGRAMA) but in the data set description link provided by the authors in the manuscript, if I understood correctly, 12 providers are listed (which, according to descriptor "dc.relation.isbasedon" are: Spanish Meteorolog- ical Agency (AEMET); Ministry of Agriculture and Environment; Servei Meteorològic de Catalunya (METEOCAT); Navarra Government; SAIH Cantábrico; SAIH Duero; SAIH Ebro; SAIH Guadalquivir; SAIH Hidrosur; SAIH Júcar; SAIH Miño-Sil; SAIH Segura; SAIH Tajo), so that implies 10 sources more than those mentioned originally in the manuscript. Obviously one of the two descriptions is not correct so this should be clarified and corrected.**

> You are right, the description in the repository is wrong. A new version of the dataset has been uploaded and now the information is correct. Thank you for the notice.

**Finally, there are a number of formal aspects that should be amended such as the use of correct symbol units, references quoted in the text but not listed in the references section, or minor problems with English language (I do not intend to be exhaustive in this aspect**

**but I list some issues below). For all the above, despite this is a very interesting contribution, I do not think the manuscript can be accepted in its current form and I recommend major revision.**

We are grateful for your comprehensive review that gave us the opportunity to solve several formal aspects. We have addressed all your suggestions and the text is now much better understandable.

**SPECIFIC COMMENTS**

**1. Page 2, line 20. Please add here the horizontal resolution of the E-OBS dataset, i.e. "with a horizontal resolution of .." or something similar.**

Modified as suggested. We added the spatial resolution of the dataset to the text.

*"The E-OBS dataset (Cornes et al., 2018), at a maximum spatial resolution of 0.1 degree, […]"*

**2. Page 2, lines 23. The first part and last parts of the sentence "The Spanish territory perfectly captures... this variability" is unclear and potentially misleading... what do you mean by "the great climatic variability"? The climatic variability of the world? I suggest to rewrite it, for example "The Spanish territory exhibits a great climatic variability with very different regimes in a relatively small area that leads to high risks such as.." (please complete with what you think are those risks) or something similar.**

Thank you. We have modified the text based on your suggestion and, as mentioned by Reviewer #1, we have included the explanation of the risks part.

*"The Spanish territory exhibits a great climatic variability with very different regimes in a relatively small area that leads to high risks such as increments in the frequency and magnitude of extreme events."*

**3. Page 3, line 13. Please complete the sentence with all data sources (as mentioned earlier) if there are more than two - or correct the 12 data sources given in the data set description repository.**

Modified as suggested.

**4. Page 3, figure 1. Map bottom panel of Canary Islands: please enlarge axis font sizes they can hardly be read**

Modified as suggested.

**5. Page 4, line 2. Please give a range of mean distances between stations, as it is done later for mean station elevations.**

We added the figure of mean distance (65.9 km) in the corresponding line.

**6. Page 4, Figure 2. This is a very important part of the data set description, as it gives an idea of temporal changes in mean altitude, minimum distance and number of stations. The methodology used by the authors allows the use of a variable number of stations, not constant in time, which is good because it maximizes the information introduced in the estimation. However, this introduces variability due to the changes in the station data set so this should be carefully described. Authors chose to show mean values only, which I think it is rather limited, and combined in the bottom panel two variables. I suggest to consider a three panel figure, one for each of the three variables considered, and to plot for each one the median and percentiles 25 and 75 - alternatively, if variables examined follow a Gaussian distribution mean values plus standard deviation could be also a possibility - however I would favour the first option. Then the new figure should be briefly**

**commented, and in particular, justify properly (here or in section 4.1) the selection of the 5 km x 5 km horizontal resolution, which in the current version seems arbitrary in this section (in the discussion the reader founds that it is consistent with an already existing precipitation data set for the same region, but this is not either mentioned in section 2 and could be also considered).**

Thank you for your suggestion. We changed Figure 2 to show in three panels the annual median altitude, distance between stations and number of stations, respectively. We also added shadings representing the intervals between 25th and 75th percentiles. Moreover, we added a comment on this new figure and a justification of the selection of the spatial resolution of the gridded dataset:

*"We used a 5 x 5 km regular grid covering the whole peninsular Spain, Balearic and Canary Islands to estimate maximum and minimum temperature values from the quality-controlled and serially-complete original series. Despite the differences in the data availability through time, the methodological process creates spatial references that are standardized with the temporal structure of the series to avoid biases or incoherencies. In this regard, the chosen spatial resolution accurately reflects the local characteristics of daily temperature in most of the temporal period, while the provided uncertainty values help to understand the reliability of the estimates when the original data have higher variability."*

**7. Page 5, section 3.1. Please introduce properly the meaning of TMAX and TMIN.**

Modified as suggested, we added the definition of these two variables in the text.

**8. Page 10, line 25. How many iterations of the procedure described are typically required?**

The number of iterations depends on the quality of the data and the size of the dataset. We added a reference to Figure S6 where the number of iterations for the Spanish dataset is recorded.

**9. Page 11, line 17. Pearson -> Pearson correlation coefficient ? [at least the first time you mention it]**

Thank you. We added your suggestion in the text.

**10. Page 14, section 4.1. Please justify or just comment briefly (if justified previously) the selection of the horizontal resolution.**

We added, as suggested, a comment about the selection of the spatial resolution of the grid at the end of section 2.

**11. Page 14, line 7. Suggest: case -> case using a leave-one out cross validation (LOO-CV) [this is mentioned later but not in the text]**

Modified as suggested. Thank you.

**12. Page 14, line 9. I think there is a rounding problem here and 0.95 should be 0.96 according to figure 7c - please check.**

You're right, thank you. Corrected.

**13. Page 14, figure 7. It is good the authors chose to show the four panels with the same x-axis and y-axis ranges to compare them properly. However, the panels are shown as rectangles and not as squares thus using a different scale for x and y axis. Could you please fix this?**

Thank you for the suggestion. The Figure has been modified to show the same X and Y axis scales and all the panels are now squares.

**14. Page 14, figure 7. Please indicate (in the figure caption is fine) the meaning of the dashed lines - confidence intervals perhaps? In that case specify the level.**

We added the explanation in the caption:

*"[…] Dashed lines represent ±1 standard deviation of the data."*

**15. Table 2 (and 3). Please use correct units where needed.**

Modified as suggested. We added the units in the caption of the tables.

**16. Table 2 title. When enumerated the variables displayed on the table, the last one is "Range: minimum and maximum..." but on the last row, first column, it appears "Pearson". Please correct. Note that including the units, as requested above, may help detect these problems.**

Thank you. Corrected.

**17. Table 2 (and 3). When you say decimal places you mean decimal digits? Please check.**

Modified as suggested.

**18. Table 3. Suggest adding in the first row, first column, the label "Altitude (m)" , referred to the values listed on the first row, next columns.**

Modified as suggested.

**19. Page 18, Figure 11. This is a very important part of the study and I think it deserves more attention. The uncertainty (upper panel) initially increases, ca. from 1900 to 1905, and then decreases from 1975. However, if I understood correctly, authors mention only a decreasing trend (page 17, line 10). Could you please clarify this? I suggest adding a background grid to the figure to allow an easier visual analysis.**

Thank you for the suggestion. The text explains that the initial increase of uncertainty in the first years is due to the few available stations. We modified the paragraph to clearly explain this part.

*"The uncertainty of the estimates showed a decreasing temporal evolution (Figure 11a) from the 1960s, while a positive trend was found in the first half of the period, especially in the first 15 years, coinciding with the moment of less observations and higher distance between them (see Figure 2b, c)."*

The figure has been also modified to show a background grid and now the uncertainty values are represented by boxplots to better show the variability.

**20. Page 18, Figure 11. I think that, given the variability of terrain heights in Spanish territory and different station densities at different altitudes, it is necessary to stratify Figure 11 into an additional 6 panel figure, considering to split the aggregated uncertainty and mean error values into different station altitudes. Looking at Table 3 altitude classes, probably 3 station groups, for example those with altitudes contained in the following intervals [0, 500), [500, 1500), [1500, ) m above sea level, would be enough. Future studies examining aspects at different terrain heights may largely benefit from this additional figure to better interpret subsequent results.**

That is a good suggestion. Based on your comments, we created a new figure that we added to the supplemental file (Figure S9). This figure represents, with boxplots, the annual evolution of uncertainty divided in three altitudinal ranges (< 500 m.a.s.l.; 500 to 1,500 m.a.s.l.; and > 1,500 m.a.s.l.).

**21. Results given in sections 4.3 and 4.4 refer both to Figure 12. I suggest to split that figure into two figures (first and last four panels respectively) so that it is easier to read the comments referred to each part of the figure.**

Modified as suggested.

**22. Regarding Figure 12f and 12h I noted that the uncertainty values (expressed in days) over the islands (both Balearic and Canary Islands) are either very low (Figure 12f) -except for the highest terrains in Canary islands - or very high (Figure 12h). Could you please comment this result?**

We added specific explanations to these points in the text to show that the high uncertainty values are due to the high variability of temperature between stations in the islands.

**Technical Comments & Minor Details**

**23. Page 1, line 24. Typo?: team -> teams? Please check meaning and correct if necessary.**

Modified as suggested.

**24. Page 1, line 26. Jones et al 2010: reference quoted but not listed in references section. Is it perhaps Jones et al 2012?**

You're right. Modified as suggested.

**25. Page 1, line 26. Please check citation journal style: Willmott and Matsura (1900- 2014) (2001) -> Willmott and Matsura (2001) ?**

Modified as suggested.

**26. Page 1, line 29. Check citation journal style: 2015 and 2018 -> 2015; 2018?**

Modified as suggested.

**27. Page 2, line 13 and line 15. "e.g.:" -> "e.g." as in line 9, same page?**

Modified as suggested.

**28. Page 2, line 27. English: did not considered -> did not consider**

Modified as suggested.

**29. Page 3, line 12. check: down -> bottom map panel**

Modified as suggested.

**30. Page 3, line 12. Aemet -> AEMET**

Modified as suggested.

**31. Page 3, line 18. English, suggest: this moment -> then [or "that moment"]**

Modified as suggested.

**32. Page 5, Figure 3 caption. Suggest: RV -> Reference Values (RV) [I know it is already defined in the text, but this change improves the readability of the figure]**

Modified as suggested.

**33. Page 5, section 3.1. Please use correct Celsius degree symbols as you have done elsewhere in the manuscript.**

Sorry that was a mistake produced by the font type. Corrected.

**34. Page 5, line 7 (and elsewhere in the manuscript). English: please check the meaning of suspected, suspect and suspicious and use properly.**

Corrected in the whole manuscript.

**35. Page 8, line 8. English: supplemental -> supplemental material**

Modified as suggested.

**36. Page 9, line 1. English: finish -> finishes**

Modified as suggested.

**37. Page 10. For some reason, in this page, text before equations end with a "." and not ":" as it is done elsewhere in the manuscript - please check.**

Modified as suggested.

**38. Page 11, line 2. English, suggest check: obtain -> obtaining**

Modified as suggested.

**39. Page 12, line 11. English: "fulfil" is correctly spelled, but it is in the British form - being the American form "fulfill". As you use previously in the manuscript "neighbor" (American form) this is not consistent: you should chose either American or British forms, but not a mixture.**

Modified as suggested.

**40. Page 13, line 5. English: what -> which? Please check and correct if necessary.**

Modified as suggested.

**41. Page 13, Figure caption 6. upper (bottom) line -> upper (bottom) row**

Modified as suggested.

**42. Page 16, line 14 (and elsewhere in text): a X% -> X% (remove "a" if only values are given)**

Modified as suggested.

**43. Page 16, line 15. English: slightly -> slight [as in line 17 same page]**

Modified as suggested.

**44. Page 17, line 2. Please expand "approx."**

Modified as suggested.

**45. Page 18, line 13. Sentence "The northern half..", please check English - by "than" you mean "to"?, otherwise compared to what?**

Modified as suggested.

**46. Page 18, last line. higher -> highest?**

Modified as suggested.

**47. Page 21, line 14. Hofstra et al 2008: reference quoted but not listed in the references section. Idem in same page references Jarvis et al 2001, Jones et al 2010 (is it 2012?), Kurtzmann and Kadmon 1999 (is it 2009?) (next page) Hubbard et al You et al ... please check carefully and make necessary corrections.**

Thank you for your comment, we have carefully checked the references and now all of them are listed in the references section.

**48. Page 21, line 11. English: lies > lied?**

Modified as suggested.

**49. Page 21, line 14 English: comes -> come**

Modified as suggested.

**50. References: please check alphabetical order - the last one (Van Den ... ) should not be there.**

Modified as suggested.

**RESPONSE LETTER**

Review of Manuscript No.: essd-2019-52
Title: STEAD: A high-resolution daily gridded temperature dataset for Spain
Author(s): Serrano-Notivoli R, Beguería S, de Luis M

**GENERAL COMMENTS**

**Overall this is a thorough piece of research, and the content is entirely suitable for this journal. The gridding method, including the quality control procedure, are well documented, but some clarifications are necessary prior to publication.**

> Thank you for your comments. We tried to be clear in the description of the methodological process and the resulting products. We hope that the changes made in the manuscript are fair enough for a straightforward understanding.

**SPECIFIC COMMENTS**

**A common problem with datasets that grid tmax and tmin separately is that there is no guarantee that tmax will be greater than tmin in the final dataset. I have checked the dataset and there are several days where tmax<tmin values occur in certain grid cells. These mostly occur across the edges of the gridding domain. The highest frequency (11) of tmax<tmin is during the year 1996, although such occurrences are apparent for most years. I do not advocate changing the methodology to account for this, but this limitation needs to be highlighted in the paper.**

> Thank you for checking carefully the dataset, we appreciate the concern. We realized that the available dataset at the *digitalCSIC* repository is a previous version rather than the final one. We have uploaded the correct version, which was used to make the analysis in the manuscript. Sorry for the mistake. The methodological process creates new estimates for maximum and minimum separately. However, all the stages, from the quality control to the gridding, consider the differences between them to regularly check the internal consistency, so it is impossible to get situations in which tmax<tmin.

**The methodology is broadly similar to the method used in the SPREAD precipitation dataset, produced by the same authors. I refer to the use of Reference Values and Generalized Linear Models, as for precipitation the skewed nature of the data and zero-cutoff were also taken into account. Nonetheless, given that there is a connection between STEAD and SPREAD I think that the precipitation dataset needs to be mentioned earlier in the introduction, and in the Methods Section the differences in the method used here for the temperature variables should be indicated.**

> Based on your suggestion, we added an extended explanation of SPREAD-STEAD relationship in the introduction:
>
> *"The experience acquired in the SPREAD dataset (Serrano-Notivoli et al., 2017a) development set the basis for a solid and reliable daily gridded precipitation datasets creation. Using the same framework with a complete renewal of the core calculations, we developed a new methodology for daily temperature datasets reconstruction and grids creation."*
>
> and also at the beginning of the methods section:
>
> *"The key stages of the methodological process (calculation of RV, quality control, gap filling and gridding) are the same to that used to create the SPREAD dataset (Serrano-Notivoli et al., 2017a). However, the method basics are completely different since the RV creation has been refined, the quality control has been adapted to temperature data, and*

*the gap filling and gridding processes include now an improved standardization procedure."*

**Section 2: Information needs to be provided about the time schedule over which the daily maximum and minimum values were recorded. This may not be available for all stations but where available it should be described briefly in this section, e.g. are tmax/tmin calculated over the full 24-hour period and does this change over time.**

Since we used the daily products provided by AEMET and MAGRAMA, we don't have the information about the moment of the day in which maximum and minimum temperatures were recorded. Anyway, this issue doesn't affect to the temperature estimates because the method creates a prediction for each original observation, independently of when it was recorded. The final reconstructed series faithfully represents the temporal structure of the original ones, regardless of the moment of recording.

We added a sentence in section 2 explaining that we used the daily maximum and minimum values of temperature observations:

*"Daily maximum and minimum values of temperatures series were used from all the observatories."*

**Section 2: As pointed out by reviewer #2, the changing number of input stations over time can have a profound influence on the gridded data, and is important for users who want to calculate long-term trends from the data to be aware of this. This limitation of the dataset needs to be highlighted.**

We contributed to the minimization of the impact of the data availability over the time introducing a standardization procedure. The methodological approach creates spatial references that are standardized with the temporal structure of the series to avoid biases or incoherencies. Furthermore, the provided uncertainty values for each of the estimates inform about the reliability of the data. This was already mentioned in the discussion section (6[th] paragraph). However, we emphasized this interesting subject in section 2:

*"[…] Despite the differences in the data availability through time, the methodological process creates spatial references that are standardized with the temporal structure of the series to avoid biases or incoherencies. In this regard, the chosen spatial resolution accurately reflects the local characteristics of daily temperature in most of the temporal period, while the provided uncertainty values help to understand the reliability of the estimates when the original data have higher variability."*

**Section 4.2: The verb "depurate" appears to me to be wrongly used for this procedure. Suggest changing to simply "Quality-controlled dataset".**

Modified as suggested.

**Section 5 (Discussion) lines 20-23: The key point about producing these gridded datasets is that the final values should reflect grid-box average values that are based on limited spatial sampling (unless the method produces values representative of point- values, which I understand that it does not). This relates to the comments by reviewer #2 about the choice of gridding resolution. This general aim of gridding is not articulated well in this discussions section and needs revision.**

Whereas your comment is very interesting, maybe there is a misunderstanding at this point. The temperature estimates (as well as their corresponding uncertainty values) are created for specific individual locations represented by 4 parameters (i.e. latitude, longitude, altitude and distance to the coast). The representativeness of the grid-box in each case falls on that each of those parameters are the median value of all covering that area. For instance, in the STEAD dataset, each gridpoint is the centroid of a squared area of 5 x 5 km with the median of all the possible values of the parameters covering that area.

We added an explanation in the description of the grid (section 2) to avoid misunderstandings:

*"[…] The predictor parameters (i.e. latitude, longitude, altitude and distance to the coast) for each grid point were computed as the median of all the possible values of those parameters, covering an area of 5 x 5 squared km in which the grid point is the centroid. […]"*

**Abstract and Conclusions: One of the key aspects of this paper is the use of many more stations than used in other datasets for the region. This needs to be stated more clearly in the abstract and conclusions, as the phrase "full total of available 5520 observatories" does not convey to the reader (especially those not familiar with the station density across the region) that this is an important feature of this dataset.**

Despite the full total available number of stations is 5,520, this figure is for the whole period, so the station density has greatly varied though time. However, we included in the abstract and conclusion, as suggested, the theoretical density:

[revised manuscript text omitted]
 by altitudinal ranges: (a) gridpoints lower than 500 m.a.s.l.; (b) gridpoints between 500 and 1,500 m.a.s.l.; (c) gridpoints higher than 1,500 m.a.s.l.**